# The transcription factor Foxp1 preserves integrity of an active Foxp3 locus in extrathymic Treg cells

Sayantani Ghosh[1,2], Sinchita Roy-Chowdhuri[3], Keunsoo Kang[4], Sin-Hyeog Im [1,2] & Dipayan Rudra[1,2]

Regulatory T (Treg) cells, which are broadly classified as thymically derived (tTreg) or extrathymically induced (iTreg), suppress immune responses and display stringent dependence to the transcription factor Foxp3. However precise understanding of molecular events that promote and preserve Foxp3 expression in Treg cells is still evolving. Here we show that Foxp1, a forkhead transcription factor and a sibling family member of Foxp3, is essential for sustaining optimal expression of Foxp3 specifically in iTreg cells. Deletion of *Foxp1* renders iTreg cells to gradually lose Foxp3, resulting in dramatically reduced Nrp1⁻Helios⁻ iTreg compartment as well as augmented intestinal inflammation in aged mice. Our finding underscores a mechanistic module in which evolutionarily related transcription factors establish a molecular program to ensure efficient immune homeostasis. Furthermore, it provides a novel target that can be potentially modulated to exclusively reinforce iTreg stability keeping their thymic counterpart unperturbed.

[1] Academy of Immunology and Microbiology, Institute for Basic Science (IBS), Pohang 37673, Republic of Korea. [2] Division of Integrative Biosciences and Biotechnology, Pohang University of Science and Technology, Pohang 37673, Republic of Korea. [3] Department of Pathology, The University of Texas MD Anderson Cancer Center, Houston, TX 77030, USA. [4] Department of Microbiology, College of Natural Sciences, Dankook University, Cheonan 31116, Republic of Korea. Correspondence and requests for materials should be addressed to D.R. (email: rudrad@ibs.re.kr)

Regulatory T (Treg) cells represent a unique subtype of CD4$^+$ T cells critical for maintaining immune homeostasis. The X-chromosome encoded transcription factor Foxp3 is a hallmark of Treg cells, whose continuous and stable expression is responsible for establishing and maintaining a unique transcriptional program that functionally and phenotypically distinguishes them from other T cell lineages[1–4]. In the past several years, research based on biochemical, genetic as well as cellular immunological experiments have firmly established that, while the major source of Treg cells within the vertebrae immune system are thymically generated (tTreg) cells, a sizable percentage of Foxp3$^+$ Treg cells are generated extrathymically from naive Foxp3$^-$ T cells as induced Treg (iTreg) cells[5,6]. In vivo, iTreg cells are preferentially generated in mucosal barrier sites such as the gut-associated lymphoid tissues (GALT), where they serve a non-redundant role in establishing and maintenance of tolerance from overenthusiastic immune response originating from gut-resident microbiota and food-derived foreign antigens[7–9]. In iTreg cells, Foxp3 expression initiates in response to T cell receptor stimulation coupled with environmental cues involving transforming growth factor (TGF)-β and interleukin 2 (IL-2) signaling, which eventually converge to a set of well-defined conserved non-coding sequences (CNSs) on the Foxp3 locus through Smad2/3 and Stat5 signaling pathways, respectively[10–13].

In recent years, Foxp1, a related transcription factor of the fork-head family, has emerged as an essential regulator of a varied range of biological processes. In particular, within the immune system Foxp1 has been implicated in negative regulation of monocyte differentiation and macrophage function[14]. Its efficient downregulation is essential for optimal germinal center B cell maturation by antagonizing the function of the transcription factor Bcl6[15]. Within the T cell compartment, Foxp1 is found to be important for maintenance of quiescence in CD4$^+$ and CD8$^+$ conventional T cells by repressing IL-7Rα expression and dampening Erk signaling[16,17]. Foxp1-deficient CD4$^+$ or CD8$^+$ T cells in the periphery spontaneously acquire an activated phenotype associated with enhanced proliferation, albeit with increased apoptosis[16]. By directly inhibiting IL-21 expression and limiting inducible T-cell co-stimulator (ICOS) expression, Foxp1 also suppresses follicular T helper cell differentiation and reduce germinal center reaction[18].

More recently, it was demonstrated that, in tumor microenvironment, TGF-β-mediated upregulation of Foxp1 primarily in CD8$^+$ T cells renders them unresponsive toward immunity against tumors. Accordingly, Foxp1-deficient lymphocytes facilitated enhanced tumor rejection and promoted protection against tumor re-challenge. Under these conditions, Foxp1 acts as an integral part of the Smad signaling pathway by interacting with Smad2 and Smad3 in a TGF-β-dependent manner[19].

Owing to this recently established connection between TGF-β signaling and regulation of Foxp1's transcriptional activity, here we investigate whether Foxp1 is an essential link between TGF-β signaling and the iTreg differentiation process and find that Foxp1, by being readily associated with the Foxp3 locus in a TGF-β-dependent manner, is critically required during multiple phases of iTreg development and maturity. Using an inducible model of temporal deletion of Foxp1 in precursor CD4$^+$ T cells, we find that Foxp1 is required for optimum expression of Foxp3 during the onset of iTreg induction. More strikingly, even a conditional ablation of Foxp1 in iTreg cells at a later developmental time point, when high-level transcription of Foxp3 is already established, results in dramatic lineage instability. By contrast, the stability of Foxp3 expression in tTreg cells remains unaffected in the absence of Foxp1. Thus our study unravels a novel iTreg-specific evolutionarily related transcription factor-mediated molecular surveillance mechanism as a key determinant for the

optimal activity of the Foxp3 locus, essential for maintaining immunological tolerance.

## Results

**Foxp1-deleted iTregs cannot maintain stable Foxp3 expression.** In order to determine whether Foxp1 can affect the iTreg differentiation process, we performed a preliminary experiment to ask whether overexpression of Foxp1 in naive T cells (Tnv) affects the yield of iTreg cells in vivo. CD4$^+$CD62L$^{hi}$Foxp3$^{Thy1.1-}$ Tnv cells sorted from Foxp3$^{Thy1.1}$ mice, in which the Thy1.1 allele is knocked into Foxp3 locus[20], were transduced with control or a retroviral vector expressing cDNA encoding the longest form of Foxp1 (Foxp1-A)[21]. An in vivo iTreg conversion assay was performed upon adoptive transfer of the transduced cells along with allelically marked Treg cells from CD45.1$^+$Foxp3$^{GFP}$ mice to RAG1$^{-/-}$ lymphopenic hosts (Supplementary Fig. 1a, b). Indeed, mice harboring cells overexpressing Foxp1 displayed an advantage over control mice for their ability to generate iTreg cells within the large intestine lamina propria (LI-LP), a prime site of iTreg induction in vivo (Supplementary Fig. 1c)[22].

We next wanted to determine whether the absence of Foxp1 negatively affects iTreg differentiation. CD4-Cre-driven conditional deletion of Foxp1 in T cells have been previously shown to result in an abnormally activated phenotype where essentially all conventional CD4$^+$ and CD8$^+$ T cells upregulate CD44, a marker associated with T cell activation[16]. Since for this study we employed an independently generated Foxp1$^{f/f}$ mouse line (Konopacki et al., manuscript in preparation), different from that previously reported[16–18], we first generated Foxp1$^{f/f}$CD4-Cre mice to recapitulate these findings. CD4-Cre-mediated deletion of Foxp1 in T cells resulted in specific and efficient deletion of Foxp1 in all compartments of CD4$^+$ and CD8$^+$ T cells (Fig. 1a). Furthermore, almost all CD4$^+$Foxp3$^-$ and CD8$^+$ T cells in these mice displayed enhanced CD44 expression from a young age. CD62L, which is downregulated in effector memory T cells, was reduced primarily in aged mice (Fig. 1b, top and middle panels). Interestingly, CD44 expression in Treg cells remained largely unaffected (Fig. 1b, lower panel). While these observations were in agreement with earlier findings[16], the perpetual activated state of the CD4$^+$Foxp3$^-$ T cells in Foxp1$^{f/f}$CD4-Cre mice, however, precluded us from efficiently comparing them with wild-type Tnv counterparts for their ability to differentiate into iTreg cells. Hence, we reasoned that more clearly interpretable results addressing this question is likely to be manifested in experiments performed on Tnv cells isolated from mice harboring Treg-specific conditional deletion of Foxp1.

To this end, we generated Foxp1$^{f/f}$Foxp3$^{IRES-YFP-Cre}$ mice (henceforth designated as "KO," and control littermates Foxp1$^{+/+}$Foxp3$^{IRES-YFP-Cre}$ designated as "WT"; littermate Foxp1$^{fl/+}$Foxp3$^{IRES-YFP-Cre}$ mice harboring heterozygous Foxp1 locus displayed an intermediate phenotype and were not used in any of the experiments) by crossing mice harboring Foxp1$^{f/f}$ allele with Foxp3$^{IRES-YFP-Cre}$ mice expressing YFP-Cre recombinase fusion protein under the control of Foxp3 regulatory elements[23]. Treg-specific deletion of Foxp1 was confirmed in the peripheral lymphoid tissues of KO mice (Fig. 1c, top panel). Within the CD4SP thymocytes, however, ~20% Foxp3$^+$ cells still retained Foxp1, a population that was found to be predominantly CD24$^{hi}$ representing an earlier developmental stage where presumably degradation of residual Foxp1 protein was not yet complete[24] (Fig. 1c, middle and lower panels). Importantly, both WT and KO mice displayed comparable percentages of CD44$^{lo}$CD62L$^{hi}$Foxp3$^-$ Tnv populations (shown later in supplementary Fig. 9c, where phenotypic characteristics of WT and KO mice are discussed) permitting efficient purification of Tnv cells for in vivo and in vitro iTreg assays.

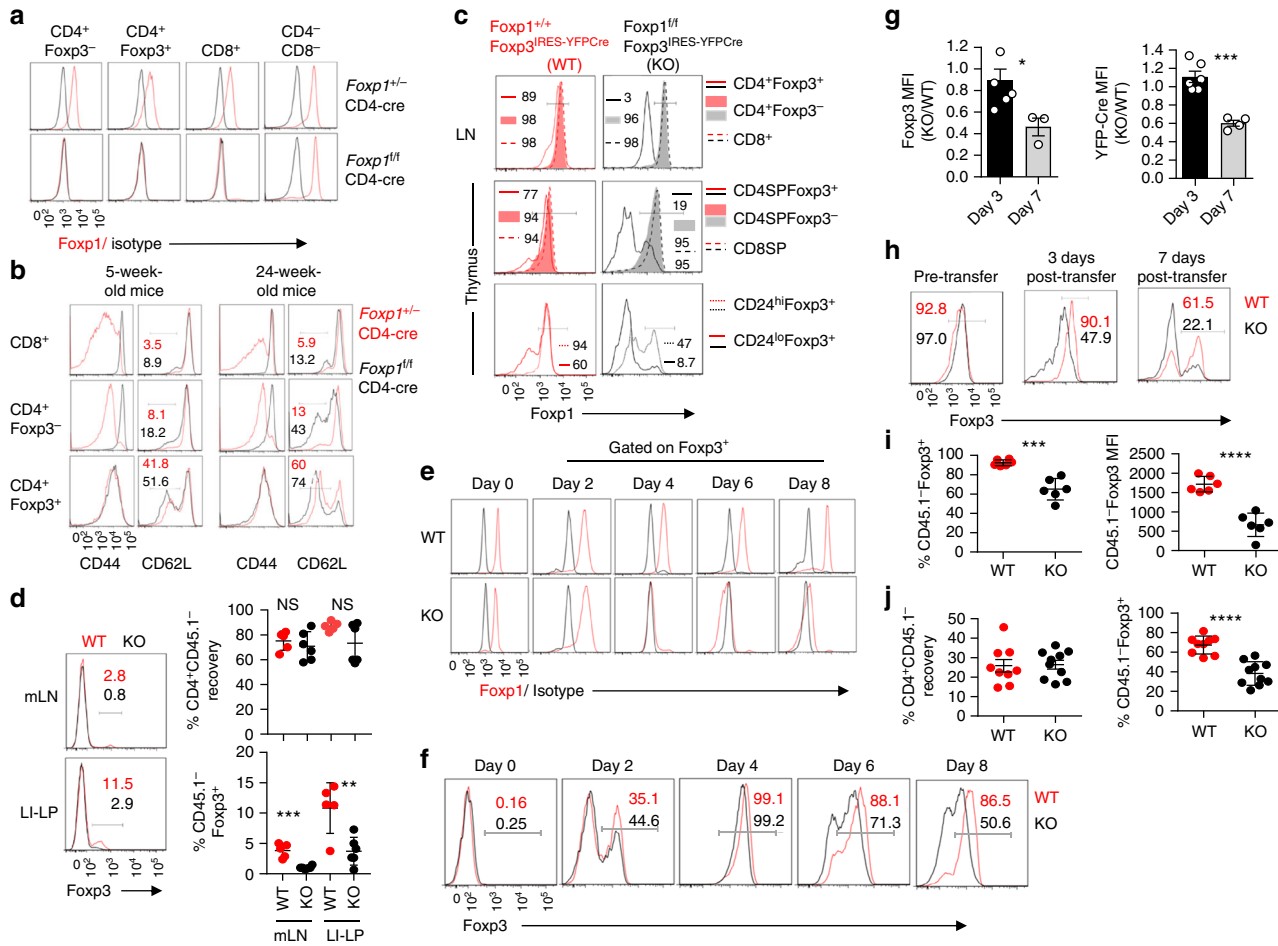

**Fig. 1** Foxp1 is required to maintain stable expression of Foxp3 in iTreg cells. **a** Intracellular staining of Foxp1 to confirm T-cell-specific deletion of Foxp1 in *Foxp1*[f/f]CD4-Cre mice. **b** Expression of CD44 or CD62L in the indicated cell populations derived from lymph nodes of young and aged *Foxp1*[+/−]CD4-Cre (red line) and *Foxp1*[f/f]CD4-Cre (black line) mice. **c** Intracellular staining to confirm Treg-specific deletion of Foxp1 in the lymph nodes (top), thymus (middle), and thymic CD24[hi]Foxp3[+] and CD24[lo]Foxp3[+] cells (bottom) of *Foxp1*[f/f]*Foxp3*[IRES-YFP-Cre] (KO) mice compared to littermate control *Foxp1*[+/+]*Foxp3*[IRES-YFP-Cre] mice (WT). **d** Representative FACS plots and quantification of percentage of in vivo iTreg conversion within CD45.2[+] compartment, 2 months after adoptive transfer of Tnv cells isolated from WT or KO mice into *RAG1*[−/−] host mice. **e** Foxp1 expression status within live CD4[+]Foxp3[+] in vitro-generated iTreg cells originating from WT- or KO-derived Tnv cells after TGF-β treatment for the indicated number of days. IgG staining is shown as isotype control. **f** Status of Foxp3 expression within CD4[+] live cell population in the same iTreg assay shown in **e**. **g** Mean fluorescent intensity (MFI) changes for Foxp3 and YFP-Cre in KO-derived iTreg cells relative to WT at the indicated time points. *$P < 0.05$, ***$P < 0.0001$ (Student's *t* test, error bars, s.e.m.). **h** WT- or KO-derived iTreg cells were sorted at high purity after 3 days of TGF-β treatment and co-transferred in *RAG1*[−/−] hosts along with allelically marked CD4[+] T cells derived from *CD45.1*[+]*Foxp3*[GFPKO] mice. Representative FACS plots of Foxp3 expression within CD4[+]CD45.2[+] compartment from lymph nodes of recipient mice at the indicated time points are shown. **i** Percentage of cells retaining Foxp3 (left) and its expression per cell depicted by MFI (right) after 3 days post-transfer. **j** Percentage of CD45.2[+] cells recovered (left) and retaining Foxp3 (right), 7 days post-transfer. Each filled circle represents a single *RAG1*[−/−] recipient mouse analyzed. Data are representative of 2–5 independent experiments. *$P < 0.05$, **$P < 0.01$, ***$P < 0.001$, ****$P < 0.0001$ (Student's *t* test, error bars, s.d.)

To directly assess a possible role of Foxp1 in iTreg differentiation, we sorted Tnv cells from WT and KO mice and adoptively transferred them in *RAG1*[−/−] hosts along with *CD45.1*[+]*Foxp3*[GFP] Treg cells. Percentage of Treg conversion within the CD45.2[+] compartment was analyzed after 8 weeks. Complementing our previous result, we observed striking reduction in the yield of iTreg population arising from cells transferred from KO donors compared to their WT counterparts (Fig. 1d).

Notably, in the Tnv compartment of KO mice at a steady state, almost all cells are *Foxp1* sufficient (Fig. 1c), which is deleted only after YFP-Cre is expressed from the *Foxp3* locus. Therefore, under this experimental setting, the transferred Tnv cells from WT or KO mice presumably expressed comparable levels of Foxp3 at an early stage of iTreg induction, which could not be

efficiently maintained upon eventual depletion of Foxp1. Indeed, a time course analysis of in vitro iTreg assay with Tnv cells isolated from WT or KO mice demonstrated this to be the case. While Foxp1 expression was indistinguishable between Tnv cells isolated from WT and KO mice at the beginning of the experiment, 4 days after TGF-β treatment, when majority of the seeded Tnv cells from both groups acquired robust Foxp3 expression, there was a complete shutdown of Foxp1 expression in KO cells (Fig. 1e and Supplementary Fig. 2a). This was accompanied by concomitant reduction in Foxp3 protein level from day 6 after TGF-β treatment, which was more apparent on day 8 (Fig. 1f). No difference in cell survival was observed between the two groups within the course of the experiment (Supplementary Fig. 2b). In-depth quantitative analyses revealed

that the relative abundance of Foxp3 protein and mRNA in KO-derived iTreg cells was reduced ~50% between 3 and 7 days after TGF-β treatment (Fig. 1g and Supplementary Fig. 2c). More strikingly, when iTreg cells of WT and KO origin, differentiated in vitro for 3 days, were sorted again and co-transferred in lymphopenic $RAG1^{-/-}$ recipients along with CD45.1+CD4+ cells isolated from Foxp3GFPKO mice (harboring Foxp3−null allele due to insertion of green fluorescent protein (GFP) in the Foxp3 locus[25]), it resulted in gradual loss of Foxp3's expression in KO-derived iTreg cells as early as 3 days after transfer. This was even more exacerbated after 7 days (Fig. 1h–j). Taken together, these results strongly suggest that a sustained high-level expression of Foxp3 in iTreg cells is dependent on the persistent presence of Foxp1.

**In vivo-generated iTreg cells require Foxp1 to remain stable.** In light of these results, we next sought to determine whether the ablation of Foxp1 could influence iTreg stability under in vivo experimental conditions. For this, we generated $Foxp1^{f/f}Foxp3^{eGFP-Cre-ERT2}R26Y$ mice in which conditional loxP-flanked Foxp1 was combined with Foxp3 allele encoding inducible Cre expressed in Treg-specific manner and a Rosa26-YFP recombination reporter allele (R26Y) enabling detection of de novo generated iTreg cells after tamoxifen administration[26]. We first ascertained the susceptibility of the $Foxp1^{f/f}$ locus for Cre-mediated recombination upon tamoxifen treatment. More than 95% of the sorted YFP+ natural Treg (nTreg) cells from these mice were found to have lost Foxp1 expression within a week after a single dose of tamoxifen, confirming high degree of recombination efficiency within the $Foxp1^{f/f}$ allele (Supplementary Fig. 3). CD4+Foxp3−CD25−YFP−CD62Lhi Tnv cells were double sorted from $Foxp1^{f/f}Foxp3^{eGFP-Cre-ERT2}R26Y$ or control $Foxp1^{+/+}Foxp3^{eGFP-Cre-ERT2}R26Y$ mice to nearly 100% purity and co-transferred along with Treg cells from CD45.1+Foxp3GFP mice into $RAG1^{-/-}$ hosts. Recipient mice were treated with tamoxifen 2 weeks after cell transfer and the fate of Foxp3 expression within YFP+ cells was analyzed after 4 weeks of treatment (Fig. 2a). The analysis involved pre-fixation of the cells with 0.5% paraformaldehyde resulting in the retention of yellow fluorescent protein (YFP) signal and flow cytometric tracking of the Foxp3+YFP+ cells generated from YFP−Foxp3− precursors[27]. As expected, tamoxifen treatment resulted in low level but comparable accumulation of YFP+ cells in both experimental groups (Fig. 2b). In order to rule out any discrepancy in the efficiency of tamoxifen-mediated Foxp1 deletion in the YFP+ iTreg cells compared to nTreg cells described above, we also confirmed efficient Foxp1 deletion within sorted YFP+ population (Fig. 2c). Indeed, under these experimental conditions, the YFP+ cells of the Foxp1-deficient origin failed to optimally retain Foxp3 expression in comparison to their Foxp1-sufficient counterparts (Fig. 2d). From this experiment, we concluded that under physiologically relevant in vivo settings iTreg cells are critically reliant on Foxp1 to preserve their acquired identity in terms of stability of Foxp3 expression.

We also considered the possibility that the enhanced loss of Foxp3 from iTreg cells derived from $Foxp1^{f/f}Foxp3^{eGFP-Cre-ERT2}R26Y$ mice might be associated with increased production of pro-inflammatory cytokines. However, the extremely small proportion of YFP+ cells generated precluded us from efficiently addressing this question by employing the above experimental set-up. In order to address this possibility, and increase the yield in YFP+ cells, we therefore employed a modified version of the experimental scheme, whereby naive T cells were briefly pulsed with TGF-β prior to transfer into $RAG1^{-/-}$ recipients. Fluorescence-activated cell sorting (FACS)-purified naive

CD4+ T cells from $Foxp1^{f/f}Foxp3^{eGFP-Cre-ERT2}R26Y$ or control $Foxp1^{+/+}Foxp3^{eGFP-Cre-ERT2}R26Y$ mice were first activated and treated with TGF-β in vitro for 24 h, following which they were transferred to RAG1-deficient hosts. After 2 days of transfer, the hosts were treated with tamoxifen and finally sacrificed for analysis after 7 days of treatment. Indeed, this modified experimental scheme yielded more than ten-fold higher proportion of YFP+ cells from both the experimental groups (compare Fig. 2b with Fig. 2e). Furthermore, we observed significantly higher frequency of Foxp3−IFN-γ+ cells within the YFP+ population derived from Foxp1-deficient group. Foxp3−IL17+ cells also showed an increased trend in frequency, although the difference remained non-significant. On the other hand, IL-4 and IL-2 production remained largely unchanged (Fig. 2f).

**Absence of Foxp1 does not affect thymic Treg stability.** The major impairment observed in iTreg cell stability in the absence of Foxp1 raised the possibility that Foxp1 might also have similar role in thymically generated tTreg cells. To test this, we performed similar adoptive transfer and fate mapping experiments with tTreg cells sorted directly from thymus. CD4SPFoxp3+ cells sorted from thymus of young WT or KO mice were co-transferred with allelically marked CD4+ T cells isolated from Foxp3GFPKO mice to $RAG1^{-/-}$ hosts and analyzed for stability of Foxp3 expression. Interestingly tTreg cells of both WT and KO origin were exceptionally stable after 7 days of transfer (Fig. 3a). As discussed earlier (Fig. 1c), while some of the KO tTreg cells retained Foxp1 expression at the time of transfer, they were confirmed to have efficiently deleted Foxp1 during the course of the experiment, suggesting that the unperturbed stability of Foxp3 in tTreg cells of KO origin is not due to incomplete deletion of Foxp1 (Fig. 3b). The observed stability of transferred cells remained unabated even 3 weeks after transfer in all intestinal tissues and peripheral lymph nodes analyzed (Supplementary Fig. 4a). Furthermore, the phenomenon was not affected by the developmental stage of the transferred tTreg cells, since both CD24hi and CD24lo tTreg populations equally retained Foxp3 (Supplementary Fig. 4b).

Finally, in order to determine whether an acute deletion of Foxp1 in tTreg cells might differentially affect Foxp3 expression, we performed fate mapping experiment where sorted thymically derived tTreg cells from $Foxp1^{f/f}Foxp3^{eGFP-Cre-ERT2}R26Y$ or control mice were co-transferred with CD45.1+Foxp3−CD62Lhi Tnv cells in $RAG1^{-/-}$ recipients. Tamoxifen was administered after 2 weeks, and mice were analyzed 4 weeks later (Fig. 3c). Similar to the previous results, tamoxifen-induced YFP+ tTreg cells from both Foxp1-sufficient or Foxp1-deficient origin displayed identical stability in Foxp3 expression (Fig. 3d). Taken together, these results indicate that, under steady-state conditions, thymically generated tTreg cells are not dependent on Foxp1 in order to maintain Foxp3 expression. Therefore the dependency on Foxp1 for sustained expression of Foxp3 is most likely an iTreg-specific phenomenon.

**Early phase of iTreg differentiation also requires Foxp1.** So far in this study, all the experiments were performed employing a conditional deletion system where Foxp1 is deleted only after Foxp3 is expressed, thereby confirming a non-redundant role of Foxp1 primarily in the maintenance of iTreg cells. Whether it also contributes during the initial stage of Foxp3's induction therefore remained undetermined. This is because to address this question the precursor T cells were required to be Foxp1-deficient, albeit being comparable to controls with respect to their activation state. As discussed earlier, this could not be achieved in $Foxp1^{f/f}$CD4-Cre mice (Fig. 1b). To circumvent this issue, we generated $Foxp1^{f/}$

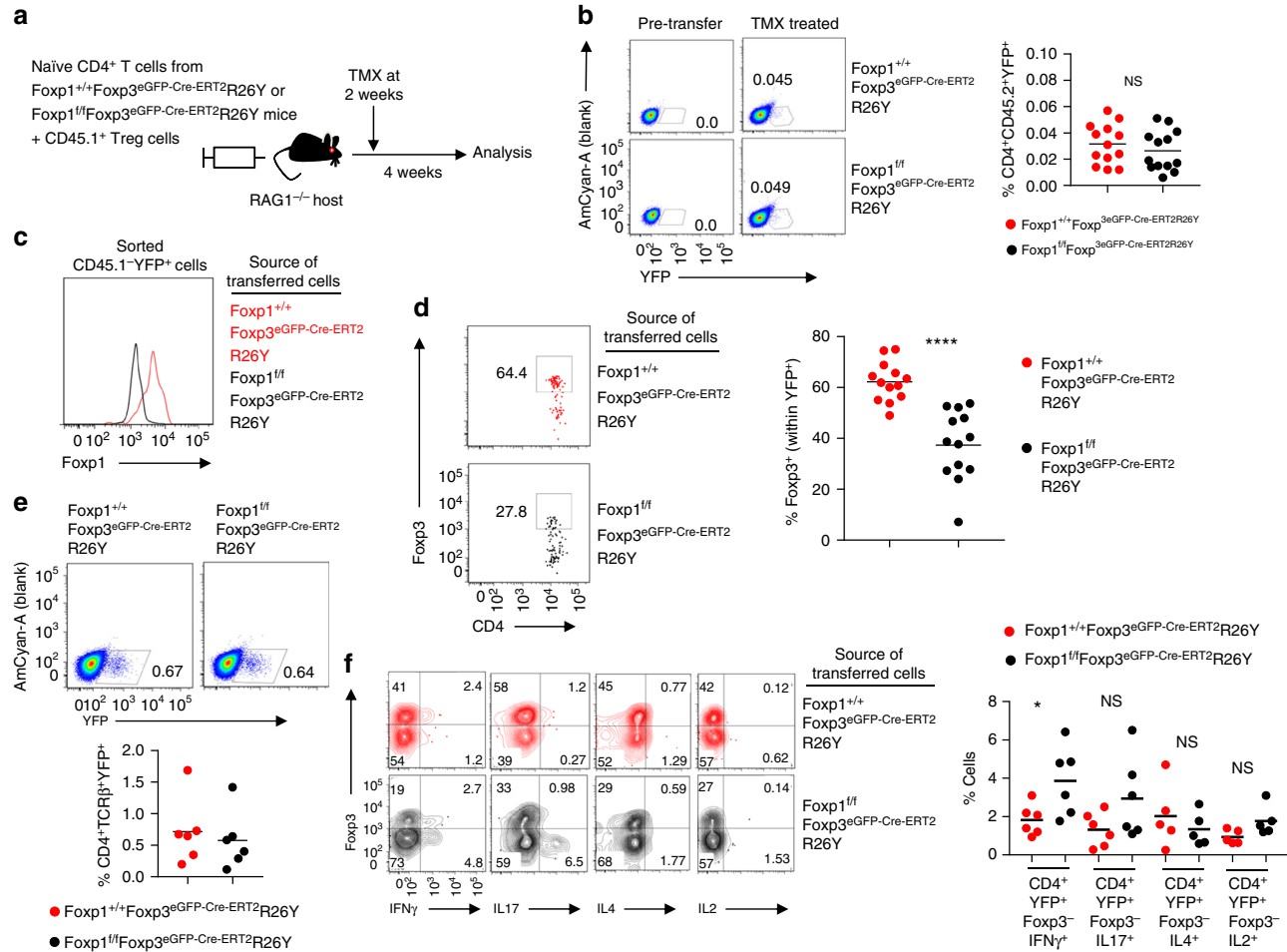

**Fig. 2** In vivo differentiated iTreg cells fail to retain stability in the absence of Foxp1. **a** Experimental scheme. Double FACS-sorted >99% pure Tnv cells from control $Foxp1^{+/+}Foxp3^{eGFP-Cre-ERT2}R26Y$ or $Foxp1^{f/f}Foxp3^{eGFP-Cre-ERT2}R26Y$ mice were co-transferred with Treg cells sorted from $CD45.1^+$ $Foxp3^{GFP}$ mice into $RAG1^{-/-}$ recipients. Mice received tamoxifen (TMX) at 2 weeks post transfer and Foxp3 expression among YFP-labeled cells was assessed after 4 weeks. **b** Four weeks after TMX treatment, individual mice were sacrificed and CD4$^+$ T cells enriched from lymph nodes and spleens were subjected to staining protocol adopted from ref. [27]. Representative FACS plots with TMX-induced YFP expression among CD4$^+$CD45.2$^+$ gate are shown. A summary of the experiment is shown at the right panel. **c** Pooled enriched CD4$^+$ cells were sorted to isolate YFP$^+$ cells and stained in order to confirm efficient deletion of Foxp1. **d** Representative FACS plots and quantification of percentage of cells retaining Foxp3 within YFP$^+$ cells shown in **b**, 4 weeks after TMX treatment. **e** Sorted Tnv cells from $Foxp1^{+/+}Foxp3^{eGFP-Cre-ERT2}R26Y$ or $Foxp1^{f/f}Foxp3^{eGFP-Cre-ERT2}R26Y$ mice were activated with plate-bound anti-CD3 and anti-CD28 and treated with TGF-β for 24 h and were transferred to $RAG1^{-/-}$ hosts, which received TMX 2 days after transfer. One week after TMX treatment, mice were sacrificed and YFP$^+$ cells were analyzed as in **b**. Representative FACS plots (top) and frequencies (bottom) of TMX-induced YFP expression among CD4$^+$ TCRβ$^+$CD45.2$^+$ gate are shown. **f** Representative FACS plots and summarized quantification depicting percentage of cytokine production from the indicated YFP$^+$ populations. Data are representative of 2–3 independent experiments. *$P < 0.05$, ****$P < 0.0001$ (Student's $t$ test, error bars, s.d.)

$^fFoxp3^{IRES-Thy1.1}Ub^{Cre-ERT2}$ mice in which deletion of Foxp1 could be achieved ubiquitously upon tamoxifen treatment. Tnv cells were sorted at high purity from $Foxp1^{f/f}Foxp3^{IRES-Thy1.1}Ub^{Cre-ERT2}$ mice, activated, and treated briefly with tamoxifen in vitro, after which it was washed out and cells were rested for 48 h (Fig. 4a). Under these conditions, we could successfully distinguish between Foxp1$^+$ and Foxp1$^-$ precursor T cells by intracellular staining (Fig. 4b). Furthermore, both the populations showed similar expression of CD44 throughout, suggesting equal activation state upon plate-bound anti-CD3/ CD28-mediated stimulation (Fig. 4c). A time course of TGF-β treatment starting at this point demonstrated that the Foxp1- deficient compartment was significantly compromised in inducing Foxp3 compared to the Foxp1-sufficient one (Fig. 4d–f). These results along with the experiments discussed earlier therefore established a bipartite role of Foxp1 during the iTreg differentiation process. At the onset of TGF-β-mediated iTreg

conversion, Foxp1 is required to ensure optimum induction, while at a later stage of development process Foxp1's involvement is critical to ensure sustained high-level expression of Foxp3.

**Foxp1 binds *Foxp3* locus to preserve its permissive state.** In addition to a promoter region, the *Foxp3* locus is comprised of well-characterized genetic elements, including three CNSs (*CNS1–3*) downstream of the transcriptional start site, which have been implicated as enhancers important for both induction and maintenance of *Foxp3*[13,28,29]. We asked whether Foxp1 associates with *Foxp3*'s regulatory elements in iTreg cells. Indeed, by performing chromatin immunoprecipitation (ChIP) assay we detected robust association of Foxp1 with the promoter and the *CNS2* regions of *Foxp3* (Fig. 5a). Importantly, the binding was dependent on TGF-β and was abolished when precursor cells are limited for Smad transcription factors (Fig. 5a, b). Furthermore,

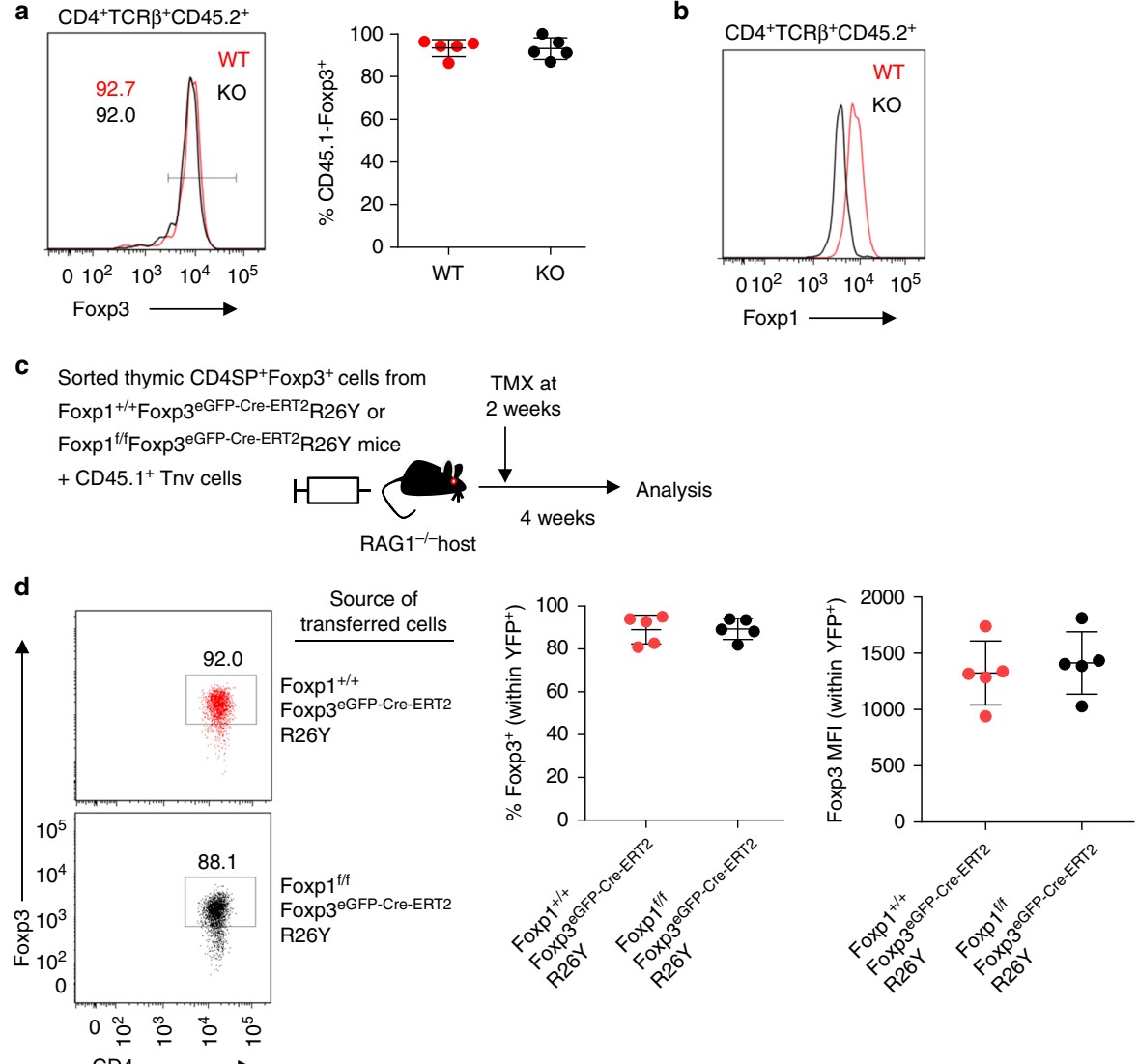

**Fig. 3** Foxp1 is dispensable for maintenance of Foxp3 expression in tTreg cells. **a** WT or KO CD4SPFoxp3⁺ tTreg cells were sorted directly from thymus at high purity and co-transferred in *RAG1⁻/⁻* hosts along with allelically marked CD4⁺ T cells derived from *CD45.1⁺Foxp3*GFPKO mice. Stability of Foxp3 expression was analyzed after 7 days post transfer. Representative histograms (left) and quantification plot (right) are shown from lymph nodes of recipient mice. **b** Representative intracellular staining showing Foxp1 protein expression in transferred WT and KO tTreg cells at the time of analysis. **c** Experimental scheme for acute depletion of Foxp1 in tTreg cells. FACS-sorted CD4SPFoxp3⁺ tTreg cells from thymus of *Foxp1⁺/⁺Foxp3*eGFP-Cre-ERT2*R26Y* or *Foxp1*f/f*Foxp3*eGFP-Cre-ERT2*R26Y* mice were co-transferred with CD4⁺ cells sorted from *CD45.1⁺ Foxp3*GFPKO mice into *RAG1⁻/⁻* recipients. Mice received tamoxifen at 2 weeks post transfer and stability of Foxp3 expression among YFP⁺ cells was assessed after 4 weeks. **d** Representative FACS plots and quantification of Foxp3 stability among the YFP⁺ cells 2 weeks post TMX treatment. Data are representative of two independent experiments

the binding was found to ensue in a stepwise manner whereby Foxp1's association with the promoter was detected first, after 24 h, followed by *CNS2* at 72 h after TGF-β treatment (Fig. 5c).

To gain insights on Foxp1-dependent molecular events responsible for conferring stability to *Foxp3* locus, we took advantage of our earlier finding that, compared to WT, the expression of Foxp3 in KO-derived in vitro-differentiated iTreg cells is ~50% downregulated after 7 days of TGF-β treatment. At this point, the effect of Foxp1 deletion appeared evident and enabled efficient harvesting of sufficiently expanded iTreg cells for molecular analyses (Fig. 1e). Since Treg cell stability has been previously correlated with cell-cycle-regulated CpG methylation status of *Foxp3* locus[28,30], particularly at the promoter and *CNS2* regions that specifically bind Foxp1, we first asked whether

instability in Foxp3 expression upon loss of Foxp1 is associated with cell division and enhanced CpG methylation. Revisiting our previous experimental schemes, we monitored cell division of iTreg cells by Cell Trace Violet (CTV) dilution alongside with expression status of Foxp3. Interestingly, upon transfer to *RAG1⁻/⁻* host both in vitro and in vivo, enhanced loss of Foxp3 in KO-derived iTreg cells were found to be independent of cell division (Supplementary Fig. 5a, b and Fig. 5d). Furthermore, bisulfite sequencing of the *Foxp3* locus of KO-derived in vitro-differentiated iTreg cells did not show any correlative increase in CpG methylation associated with reduced Foxp3, even when status of individual CpG residues were considered (Fig. 5e).

We next asked whether the state of transcription-associated chromatin modifications in *Foxp3* locus are affected in the

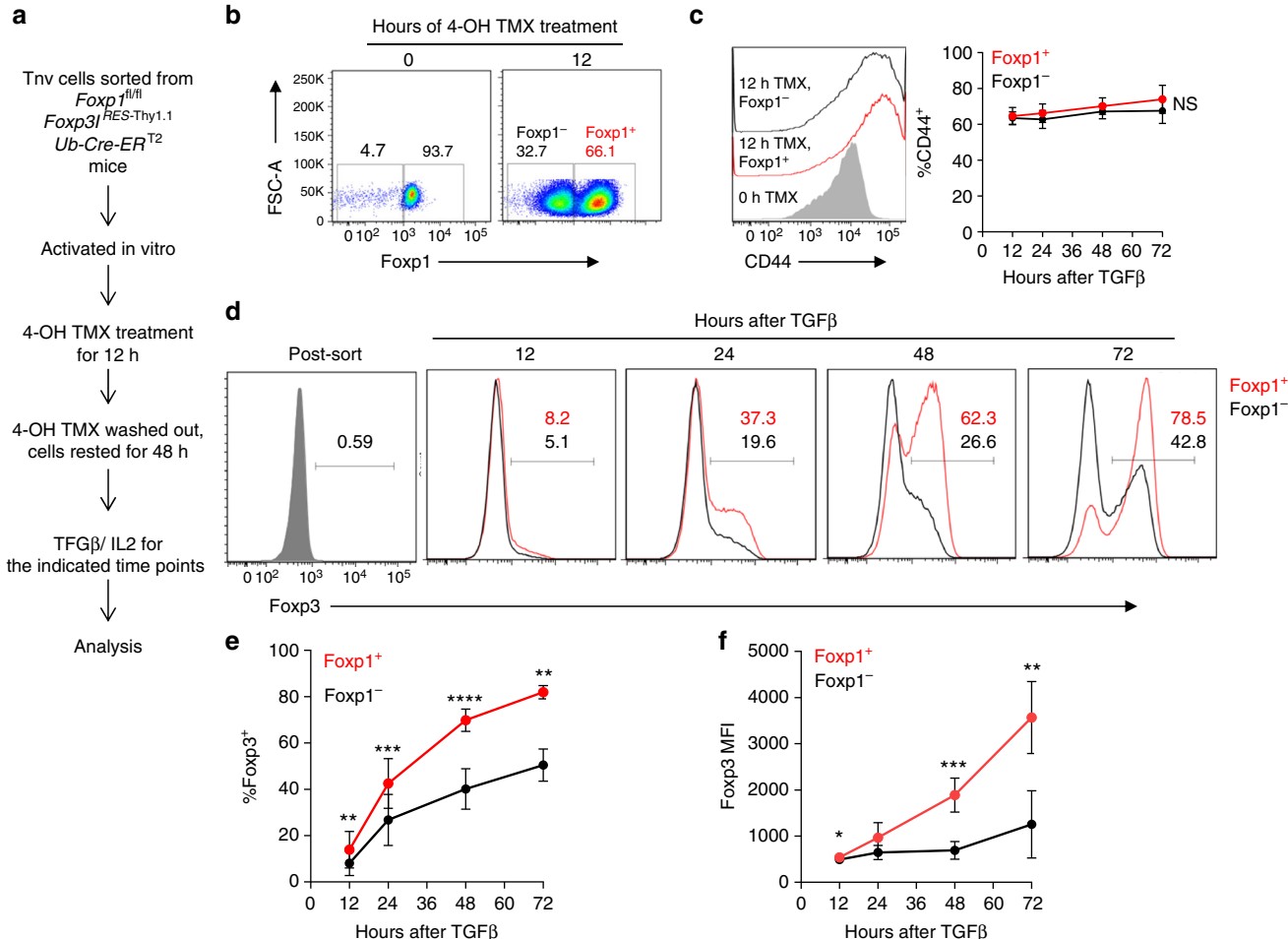

**Fig. 4** Early stage of Foxp3 induction in iTreg cells also require Foxp1. **a** Experimental scheme. FACS sorted Thy1.1⁻ Tnv cells from *Foxp1*^f/f*Foxp3*^IRES-Thy1.1*Ub*^Cre-ERT2 mice were activated in vitro with 1 µg/ml of plate-bound anti-CD3 and CD28 in the presence of 0.3 µM 4-OH tamoxifen and 50 IU/ml of IL-2. After 12 h, cells were washed, rested for 48 h, and subsequently treated with TGF-β, and the expression of Foxp3 was analyzed at the indicated time points within live Foxp1-sufficient and Foxp1-deficient cells. **b** Representative FACS plot indicating 4-OH TMX-mediated deletion of Foxp1 within a subset of cells after 48 h is shown, which were gated as Foxp1-sufficient and Foxp1-deficient populations in order to quantitate iTreg induction. **c** Representative FACS plots and quantification of CD44 expression in Foxp1⁺ and Foxp1⁻ populations throughout the course of the experiment. **d** Representative FACS plots to compare Foxp3 expression at the indicated time points between Foxp1-sufficient and Foxp1-deficient populations generated in culture after 4-OH tamoxifen treatment. **e** Quantification of percentage of Foxp3 induction. **f** Quantification of Foxp3's MFI. Data are representative of four independent experiments. *$P < 0.05$, **$P < 0.01$, ***$P < 0.001$, ****$P < 0.0001$ (Student's *t* test, error bars, s.d.)

absence of Foxp1. Indeed, iTreg cells of KO origin were found to retain dramatically lower abundance of permissive trimethylation modifications of histone H3 at lysine 4 (H3-K4me3) after 7 days in culture compared to WT (Fig. 5f, top panel). Notably, the modification was undetectable in Tnv cells and was equally abundant in both WT- and KO-derived iTreg cells in an earlier 3-day culture, at which point Foxp1 was not efficiently deleted (Supplementary Fig. 6 and Fig. 1e). Acetylated H3-K9/14 (H3-K9/K14Ac), another representative mark for permissive chromatin modification, was also found to be dramatically affected in KO-derived iTreg cells after 7 days in culture. In contrast, the inhibitory H3-K9me3 modification remained unchanged (Fig. 5f, middle and lower panels). Taken together, these results implicated an active role of Foxp1 in preserving favorable state of chromatin modifications of the *Foxp3* locus by being physically associated with its regulatory elements. On the other hand, its potential involvement in directly restoring cell-cycle-dependent CpG hypo-methylation status appears less likely.

**Compromised iTreg homeostasis in *Foxp1*^f/f*Foxp3*^YFP-Cre mice.** In order to determine the functional significance of the observed roles of Foxp1 in the iTreg differentiation process, we next performed in-depth analyses of WT and KO mice and asked whether Treg-specific deletion of Foxp1 resulted in altered iTreg homeostasis in vivo. First, upon overall characterization of the Treg cell population we observed a modest but significant reduction in the percentage of CD4⁺Foxp3⁺ Treg cells, although not in absolute numbers, particularly in peripheral lymph nodes and mesenteric lymph nodes of the KO mice compared to WT (Fig. 6a). Furthermore, the KO mice displayed enhanced Treg activation, characterized by increased percentage of CD62L^loCD44^hi and CD103⁺ Treg populations, as well as proliferation, determined by increased expression of the proliferation marker Ki67 (Fig. 6b, left panel). However, the difference in frequency of Treg cells expressing these markers do not always consistently reflect changes in the absolute number of cells (Fig. 6b, right panel).

Importantly, consistent with the results so far, we observed dramatically reduced Treg-specific expression of *Dapl1* and

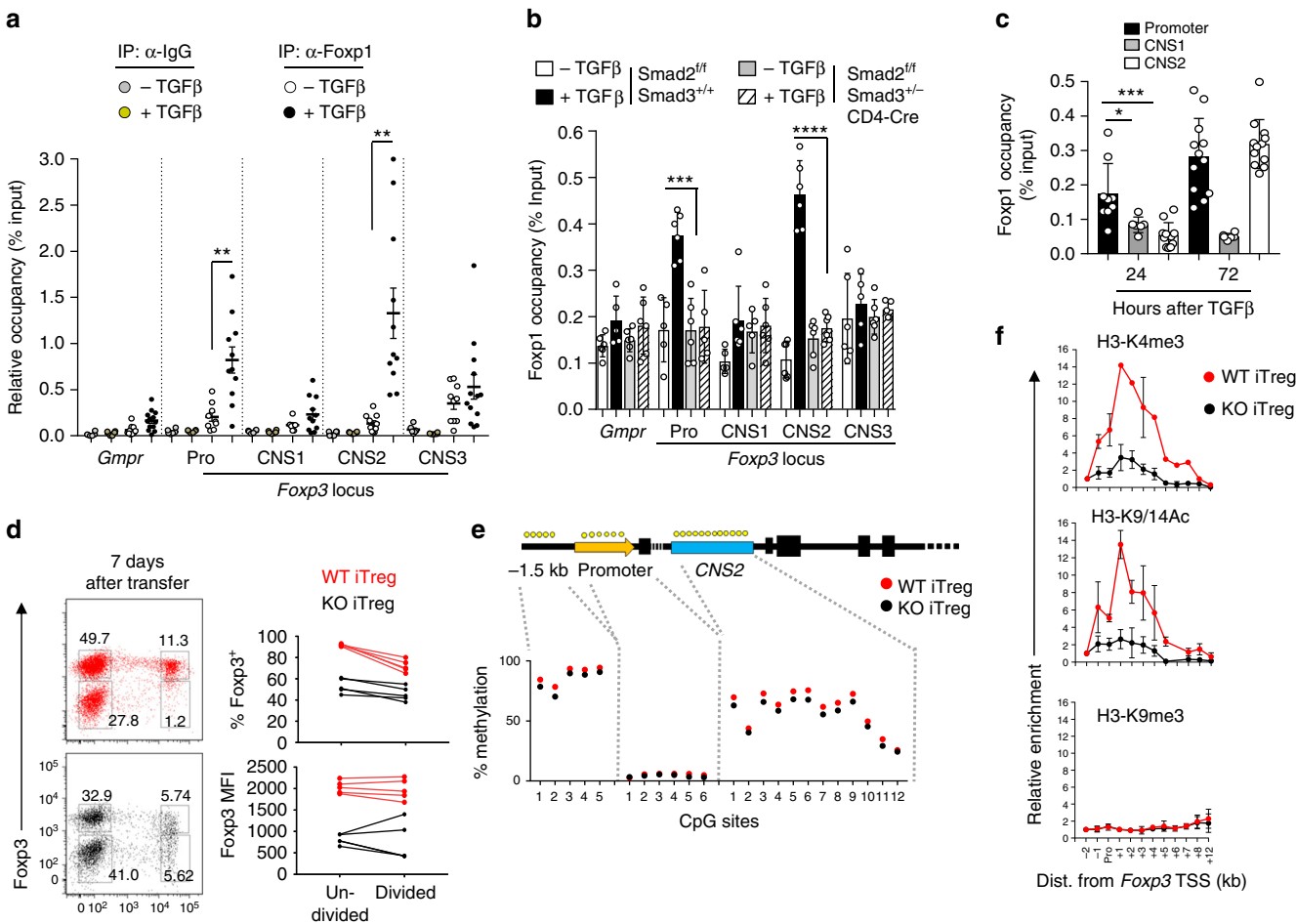

**Fig. 5** Foxp1 binds to *Foxp3* locus and retains permissive histone modifications. **a** Sorted CD4[+] Tnv cells from *Foxp3*[IRES-Thy1.1] mice were activated in vitro in the presence or absence of TGF-β for 3 days. ChIP analysis was performed with either anti-Foxp1 or normal rabbit IgG (control) antibody. Data are shown as a relative enrichment of immunoprecipitated DNA corresponding to *Foxp3* gene locus over Input. *Gmpr* serves as negative control. Data representative of at least four independent experiments. *$P < 0.05$, **$P < 0.01$, ***$P < 0.001$, ****$P < 0.0001$ (Student's $t$ test, error bars, s.e.m). **b** ChIP analysis with anti-Foxp1 antibody on iTreg cells differentiated for 3 days from CD4[+]CD25[−] Tnv cells derived from *Smad2*[f/f]*Smad3*[+/−]CD4-Cre or littermate *Smad2*[f/f]*Smad3*[+/+] mice. Data represents two independent experiments. **c** ChIP analysis to determine Foxp1 occupancy at *Foxp3* gene locus at the indicated time points. Data are representative of three independent experiments. *$P < 0.05$, **$P < 0.01$, ***$P < 0.001$, ****$P < 0.0001$ (Student's $t$ test, error bars, s.d.). **d** Sorted Tnv cells from WT and KO mice were differentiated into iTreg cells in vitro for 3 days when they were sorted, labeled with CTV, and co-transferred in *RAG1*[−/−] hosts along with allelically marked CD4[+] T cells derived from *CD45.1*[+]*Foxp3*[GFPKO] mice. Seven days post transfer, recipient mice were sacrificed and Foxp3 expression was analyzed simultaneously with CTV dilution. Representative FACS plots within CD4[+]TCRβ[+]CD45.2[+] gate and quantification of percentage of cells retaining Foxp3 and MFI of Foxp3 within the indicated populations are shown. Data are representative of two independent experiments. **e** Comparison between CpG methylation levels in the indicated regions of *Foxp3* in WT- or KO-derived iTreg cells after 7 days of differentiation. Methylated CpG levels detected by bisulfite sequencing in two biological replicate samples were averaged for each residue from >200,000 reads. Percentage of methylation of individual CpG sites is shown. **f** Relative enrichment for the permissive H3-K4me3, H3-K9/14Ac, and non-permissive H3-K9me3 chromatin modifications throughout the *Foxp3* locus in WT- or KO-derived iTreg cells after 7 days of differentiation. Relative distances (kb) from Foxp3 transcription start site (TSS) of primer probes are indicated in the *x* axis. Pro Promoter. Representative data from one of at least three independent experiments are shown

decreased frequencies of Foxp3[+]Nrp1[−] or Foxp3[+]Helios[−] Treg cells, all of which are associated markers of the iTreg population (Supplementary Fig. 7a–c)[22,31,32]. In particular, the Foxp3[+]Nrp1[−]Helios[−] population, which we anticipated to be most stringent representative of iTreg cells, was decreased specifically, both in frequency and numbers, within peripheral lymphoid organs and was most prominent in aged mice (Fig. 6c and supplementary Fig. 7d, e). These observations, along with the results so far, implicated that Treg-specific deletion of Foxp1 indeed results in reduced iTreg frequency in KO mice.

In spite of being significantly reduced, the KO mice still did harbor a small but sizable population of iTreg cells (Supplementary Fig. 7b, c and Fig. 6c). In order to characterize this population, we sorted Foxp3[+]Nrp1[lo] and Foxp3[+]Nrp1[hi] cells from KO mice and their littermate controls (Supplementary Fig. 8a) and performed real-time PCR as well as in vitro suppression assay to determine the expression status of *Foxp1* and *Foxp3* and their suppressive capacity, respectively. Interestingly, by real-time PCR analysis we observed a significant level of residual *Foxp1* mRNA still remaining in the Foxp3[+]Nrp1[lo] population derived from KO mice (Supplementary Fig. 8b). Although this level of *Foxp1* mRNA is 10-fold lower than that observed in WT Foxp3[+]Nrp1[lo] population, it is significantly higher than that observed in Nrp1[hi], for which the level of *Foxp1*

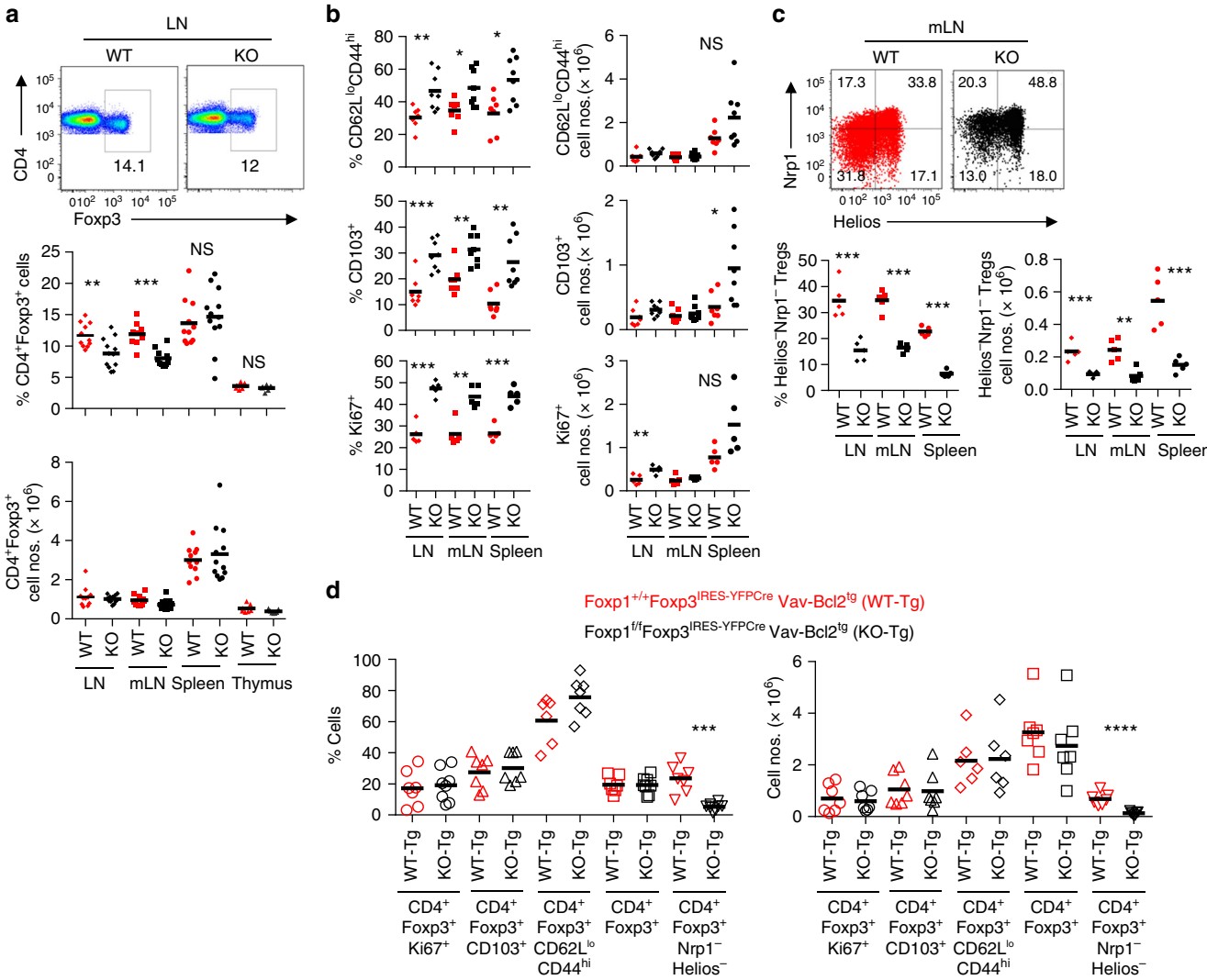

**Fig. 6** Reduced iTreg compartment in *Foxp1*f/f*Foxp3*IRES-YFP-Cre mice. **a** Representative flow cytometric analysis (top), and quantification of percentage (middle) and absolute numbers (bottom) of CD4+Foxp3+ Treg cells in the indicated organs of WT and KO mice. **b** Quantification of percentage (left) and absolute numbers (right) of Treg activation represented by CD62LloCD44hi and CD103+ and proliferation by Ki67+ cells within CD4+Foxp3+ population. **c** Representative flow cytometric analysis (top) and quantification (bottom) of in vivo Foxp3+Nrp1−Helios− iTreg population within the indicated peripheral lymphoid organs. **d** *Foxp1*f/f*Foxp3*IRES-YFP-Cre mice were bred with *Vav-Bcl2*tg mice to generate WT-Tg and KO-Tg mice. Expression of the indicated markers as percentage (left) and total numbers (right) in mLN of WT-Tg and KO-Tg are shown. Data are representative of 2–4 independent experiments. *P < 0.05, **P < 0.01, ***P < 0.001, ****P<0.0001 (Student's *t* test, error bars, s.d.)

mRNA was found to be 40-fold reduced in the KO cells compared to WT (Supplementary Fig. 8b). As expected, by intracellular FACS staining, a small fraction of these KO Foxp3+Nrp1lo cells were found to still retain Foxp1 protein (Supplementary Fig. 8c). Furthermore, the expression of Foxp3 in Nrp1lo KO Treg cells was found to be unaltered, both at the mRNA and protein level (Supplementary Fig. 8d, e). These observations suggested that the Nrp1lo Treg population in the KO mice most likely represents a small proportion of "recently generated" iTreg cells in which the process of Foxp1deletion is still ongoing, and its effect on Foxp3 expression would presumably be more apparent with time. Indeed, rendering enough time to the KO Nrp1lo Treg cells, upon culturing them in vitro for 3 days, resulted in complete deletion of Foxp1, eventually leading to instability in Foxp3's expression (Supplementary Fig. 8f, g). Finally, in an in vitro suppression assay, while the Nrp1hi populations were comparable, the Nrp1lo population derived from KO cells displayed a modest reduction

in suppressive capacity compared to their WT counterparts (Supplementary Fig. 8h).

Foxp1 deletion is frequently associated with enhanced apoptosis in several cell types[16,33,34]. Therefore, although enhanced cell death was not detected in vitro (Supplementary Fig. 2b), the reduced size of Treg compartment and heightened Treg cell activation and proliferation in KO mice raised the possibility that increased susceptibility to cellular turnover resulting from Foxp1 depletion may contribute to the observed scarcity in iTreg population in these mice. In order to address this, we crossed the *Foxp1*f/f*Foxp3*IRES-YFP-Cre mice with mice expressing hematopoietic cell-specific transgene encoding the antiapoptotic protein Bcl2 (called *Vav-Bcl2*tg and resultant mice referred to as WT-Tg and KO-Tg). Transgenic expression of Bcl2 almost completely rectified the heightened activation and proliferation as well as restored the diminished total Treg frequency in KO-Tg mice. However, these mice still remained

significantly compromised for the Foxp3[+]Nrp1[−]Helios[−] iTreg compartment, suggesting that the reduction of iTreg cells in KO mice is not due to specific increase in cell death within this population (Fig. 6d).

Subsequent analyses of the KO mice displayed moderate increase in cellularity specifically within lymph nodes compared to WT counterparts (Supplementary Fig. 9a), and while the CD4[+] and CD8[+] T cell compartments were largely unaltered (Supplementary Fig. 9b), this was associated with marginally increased frequencies of CD62L[lo]CD44[hi]Foxp3[−] effector memory population, as well as increased proportion of IFN-γ-producing effector and cytotoxic T cells within lymph nodes (Supplementary Fig. 9c, d).

**Intestinal inflammation in Foxp1[f/f]Foxp3[IRES-YFP-Cre] mice.** Since in this study we sought to focus on the phenotypes associated with impaired iTreg homeostasis, we specifically investigated the intestinal immune system associated with GALT, where we reasoned the iTreg-related phenotypes are likely to be manifested most. As in secondary lymphoid tissues, Foxp3[+]Nrp1[−]Helios[−] iTreg population was significantly reduced in Peyer's patches, small intestine, and LI-LP, the difference being of maximum magnitude in the LI-LP (Fig. 7a). The GALT-tissue-derived T cells displayed no detectable differences in terms of pro-inflammatory cytokine expression at young age (Fig. 7b, top left and middle panel, Supplementary Fig. 10 left panel). However, there were significantly increased frequencies of IL-17-producing T cells specifically within the LI-LP of aged mice (Fig. 7b, top right and lower panel, Supplementary Fig. 10 right panel). Furthermore, histopathological examination of colonic tissues derived from aged KO mice revealed submucosal thickening and crypt architectural distortion along with lymphocytic infiltration resulting in high histological score, pointing toward disruption of immune homeostasis at the mucosal surface (Fig. 7c).

In order to determine the relative contributions of impaired iTreg homeostasis versus functional incompetence of tTreg cells in gut-associated inflammation upon Foxp1 deletion, we next employed a cell transfer model of colitis. Colitogenic CD4[+]Foxp3[−]CD45RB[hi] T cells were transferred to RAG1[−/−] recipients either alone, along with total WT nTreg cells, with sorted tTreg cells from the thymus of young WT or KO mice, or with 3 days in vitro-differentiated iTreg cells derived from WT or KO mice. Of note, as shown earlier (Fig. 1h left panel) at this time point, both WT- and KO-derived iTreg cells retain comparable levels of Foxp3. As expected, severe body weight loss that developed upon CD45RB[hi] cell transfer alone could be completely prevented by total WT nTreg cells and partially by WT iTreg cells, given the non-redundant roles of thymic and induced Treg cells in preserving mucosal tolerance[35,36]. However, co-transfer of in vitro-induced KO iTreg cells failed to provide any protection as indicated by their increased body weight loss and inflamed colon (Supplementary Fig. 11a). Notably, similar result was obtained when, as an alternative approach, Tnv cells from Foxp1[f/f]Foxp3[eGFP-Cre-ERT2]R26Y or littermate Foxp1[+/+]Foxp3[eGFP-Cre-ERT2]R26Y mice were used in the presence of 4-OH tamoxifen as sources of YFP[+] iTreg cells (Supplementary Fig. 11b, c). Interestingly, the tTreg cells derived from KO mice remained largely protective, and while mice co-transferred with KO derived tTreg cells displayed a tendency of increased weight loss compared to the WT tTreg counterparts toward later time points, the difference remained non-significant throughout the course of the experiment (Supplementary Fig. 11a). Taken together, these results revealed that Treg-specific deletion of Foxp1 is moderately associated with lymphoproliferative disorder and age-associated inflammation at mucosal sites, a major cause of which can be attributed to impaired iTreg homeostasis.

Finally, the above results prompted us to investigate whether Treg-specific ablation of Foxp1 might trigger these mice to respond to dextran sodium sulfate (DSS)-induced colitis with exacerbated disease severity. Indeed, even at a young age of 6–8 weeks, chronic colitis induction with 2% DSS in KO mice resulted in increased body weight loss accompanied with epithelial degeneration, crypt destruction, immune cell infiltration as well as characteristic focal areas of squamous metaplasia, and overall heightened histological scores (Fig. 7d, e). The phenotype was accompanied with significant increase in IL-17[+] and IFN-γ[+]IL-17[+] effector T cell populations in the large intestine (Fig. 7f). Thus Treg-specific deletion of Foxp1 rendered mice with enhanced susceptibility to a physiologically relevant experimental model of colitis.

**Discussion**
Stable commitment to specific types of T cell subsets is broadly dependent on the expression of lineage-specifying transcription factors. However, potential involvement of additional regulators in establishing cell-type-specific genetic landscape for sustained maintenance of the lineage-specifying factor appears evident. Substantial effort is now being devoted to identify such crucial molecular elements amenable to be fine-tuned in order to bolster or undermine the unilateral propagation of a specific lineage in the context of health and disease. In this study, by identifying Foxp1 as a key determinant for prolonged stability of the Foxp3 locus, we significantly advance our basic understanding of the molecular events contributing to immune cell homeostasis. Furthermore, the unique dependence of iTreg cells on the presence of Foxp1 makes these findings particularly interesting since a major bottleneck in utilizing in vitro-generated iTreg cells for greater therapeutic purpose is their intrinsic unstable nature[37–40].

We found that Foxp1 can associate with the Foxp3 locus upon TGF-β treatment, first with its promoter, followed by the enhancer region CNS2. The stepwise pattern of binding, along with previous findings where Foxp1 was reported as an integral part of Smad signaling pathway in tumor-infiltrated CD8[+] T cells, strongly indicated that its initial recruitment to Foxp3 promoter is likely a Smad-dependent event. Indeed, in in vitro-generated iTreg cells where precursor T cells were derived from Smad2[f/f]Smad3[+/−]CD4-Cre mice, the binding was completely abolished. Of note, both Smad2 and Smad3 alleles were targeted for these experiments in order to minimize redundancy observed previously in Smad signaling. Furthermore, a heterozygous Smad3[+/−] allele was employed instead of Smad3[−/−], since complete deletion of both Smad alleles results in overt auto-immunity[12]. The CNS2 region is a well-established genetic element shown to be critical for maintaining Foxp3 stability in pan nTreg population[13,28,29]. The CNS2 locus is enriched for CpG sites whose methylation status largely correlates with enhanced stability of the Foxp3 locus in a cell-cycle-dependent manner[28]. Interestingly, we found that the stability-exerting function of Foxp1 on the Foxp3 locus, within the experimental conditions utilized in our study, is independent of cell division and its associated CpG methylation status. One caveat of these experiments is that under in vitro differentiation conditions the overall CpG methylation of the CNS2 region is minimally reduced even in WT iTreg cells. However, when individual CpG residues within the CNS2 region are analyzed, consistent with a previous report[41], we found that unlike other sites, few specific CpG residues, for example, residues 2, 10, 11, and 12, displayed enhanced demethylation. These observations suggest a potentially sequential nature of events in which certain key CpG residues within CNS2 are demethylated first. However, the extent of demethylation even within these specific residues remained comparable between WT-

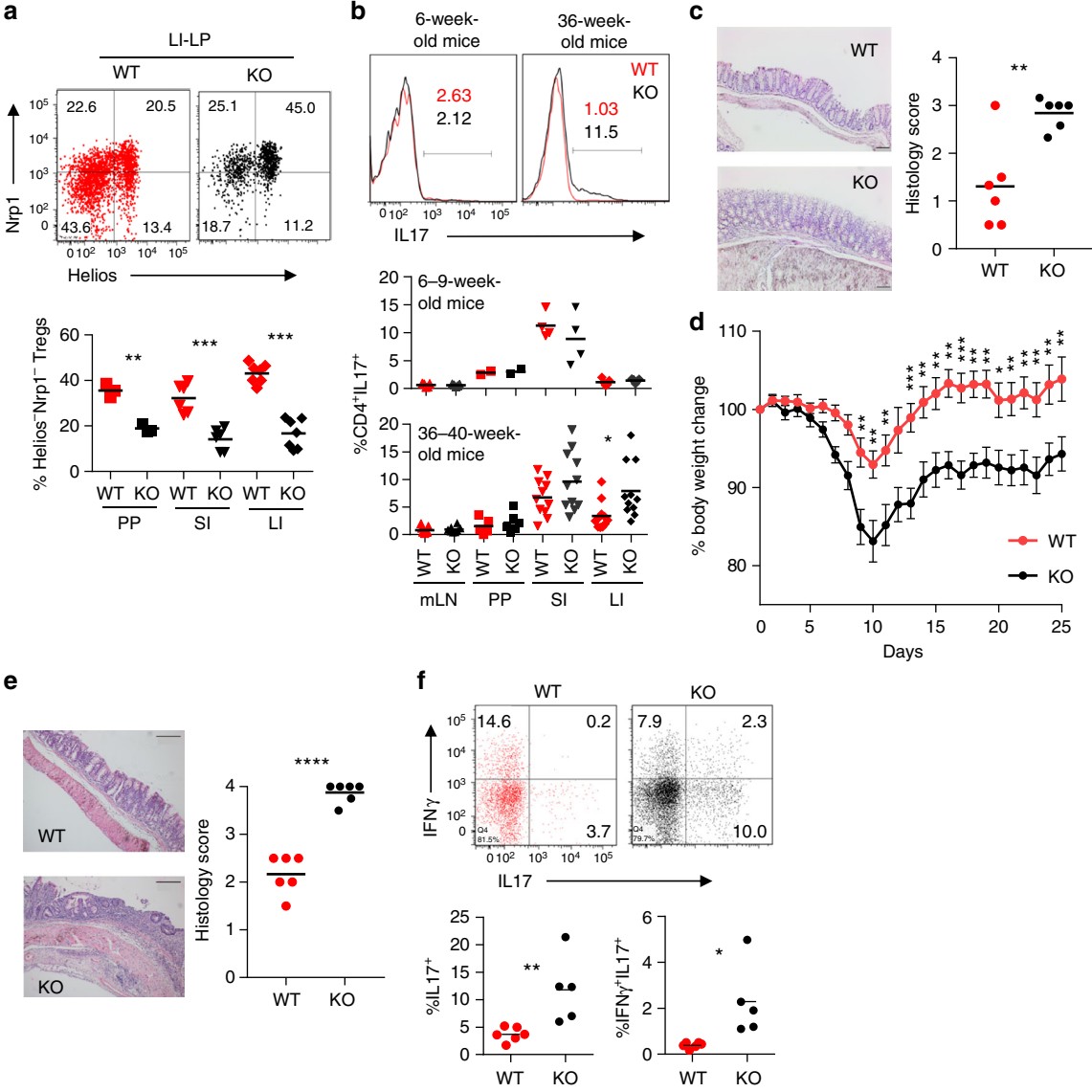

**Fig. 7** Inflammation in the intestinal mucosa of *Foxp1*<sup>f/f</sup>*Foxp3*<sup>IRES-YFP-Cre</sup> mice. **a** Representative flow cytometric analysis from LI-LP (top) and quantification (bottom) of Foxp3⁺Nrp1⁻Helios⁻ iTreg population within GALT tissues from WT and KO mice. **b** Representative flow cytometric analysis from LI-LP (top) and quantification of percentage of IL17-producing CD4⁺Foxp3⁻ cells in 6–9-week- (middle panel) and 36–40-week-old (lower panel) WT and KO mice. **c** Representative histological sections (left) and histological scoring (right) of colon from aged WT and KO mice. Sections were stained with hematoxylin and eosin. Original magnification ×100, scale bar equals 100 μm. **d** Susceptibility of WT and KO mice to 2% DSS-induced chronic model of colitis. Body weight loss over time is presented as a read out of disease severity. Two-way ANOVA with Bonferroni post-test was applied and statistically significant differences in mean values is considered at *P* < 0.05. Data are representative of two experiments (*n* = 10 for each group). **e** Histological sections (left) and scoring (right) of colon from representative DSS-treated WT and KO mice. Sections were stained with hematoxylin and eosin. Original magnification ×100, scale bar equals 100 μm. **f** Intracellular staining and quantification of colonic IFN-γ- and IL17-producing CD4⁺Foxp3⁻ cells from representative DSS-treated mice. PP Peyer's patches, SI small intestine, LI large intestine, mLN mesenteric lymph nodes. Data are representative of 2–4 independent experiments. *P < 0.05, **P < 0.01, ***P < 0.001, ****P<0.0001 (Student's *t* test, error bars, s.d.)

and KO-derived iTreg cells. Contrary to these results, deletion of Foxp1 dramatically affected permissive chromatin modifications within the *Foxp3* gene locus that correlated well with its diminished transcription. Although these results point toward an apparent disconnect between DNA hypomethylation and permissive chromatin modification, nonetheless they are in agreement with previous findings that, in TGF-β-mediated ex vivo-differentiated iTreg cells, enhanced H3K4 trimethylation appears even in the absence of DNA hypomethylation[30,42]. This is not unusual given that our understanding of precise mechanistic events contributing toward the molecular complexity of histone

remodeling and DNA demethylation, and its link with transcriptional state of gene expression is still evolving[43]. These results therefore are suggestive of multiple layers of molecular events associated with a favorable transcriptional state of *Foxp3* in iTreg cells.

Whether identical mechanisms exist during de novo differentiation of iTreg cells under physiological conditions, however, is not entirely clear due to technical constraints. Nevertheless, genetic fate mapping experiments unequivocally demonstrated that persistent expression of Foxp3 within in vivo-differentiated iTreg cells indeed is greatly dependent on Foxp1. Furthermore,

Treg-specific deletion of Foxp1 resulted in significantly reduced Nrp1⁻Helios⁻ iTreg population, which was phenotypically reflected in enhanced age-related and DSS-induced intestinal inflammation in these mice. Of note, another potential Treg-specific function of Foxp1 that was not the primary focus of this study is that it is likely to be directly contributing to Treg gene expression program by being physically associated with Foxp3[44,45]. By employing a co-transfer model of colitis induction, however, we observed that unlike Foxp1-deficient in vitro-differentiated iTreg cells, the Foxp1-deficient tTreg cells remained largely protective compared to their wild-type counterparts. This observation suggests that at least a major contributing factor for intestinal inflammation in mice bearing Treg-specific deletion of Foxp1 is impaired iTreg homeostasis.

One outstanding question is why are iTreg cells specifically affected in the absence of Foxp1? It seems reasonable to hypothesize that the differential requirement of Foxp1 to confer stability to Foxp3 locus in tTreg and iTreg cells is a direct consequence of distinct molecular features governing Foxp3's transcriptional state in these cells. Indeed, there is accumulating evidence suggesting differential roles of various transcription factors on tTreg and iTreg differentiation and functions. For example, Satb1, a global genome organizer and a regulator of transcription has been proposed to act as a pioneering factor critical for initiating Foxp3's expression in thymic Treg precursor cells. On the other hand, deletion of Satb1 in the periphery results in enhanced iTreg homeostasis[46]. The transcription factor KLF2 was reported to be crucial exclusively for iTreg induction[47], while another transcription factor musculin was recently reported to promote unidirectional stability in iTreg cells by inhibiting Th2-specific gene expression[48].

A second possibility contributing to Foxp1 dependence in iTreg cells was that, compared to tTreg cells, Foxp1 plays a more prominent antiapoptotic role in iTreg cells. As a result in Foxp1^{f/f}Foxp3^{IRES-YFP-Cre} mice, Foxp1's ablation in iTreg cells might have resulted in preferential cell death in this compartment even before Foxp3's expression is altered. However, two independent observations suggest that this is unlikely. First, in fate mapping experiments employing Foxp3^{eGFP-Cre-ERT2}R26Y mice, the frequency of tamoxifen-induced YFP⁺ cells arising from Foxp1-sufficient and Foxp1-deficient background were nearly identical even after 4 weeks post tamoxifen treatment. Second, the diminished Foxp3⁺Nrp1⁻Helios⁻ iTreg compartment in Foxp1^{f/f}Foxp3^{IRES-YFP-Cre} mice remained unaltered after overexpression of the antiapoptotic factor Bcl2.

In summary, our results demonstrate Foxp1 as a cardinal mediator of the iTreg differentiation process. We propose that such non-redundant requirement of an evolutionarily related sibling of a key lineage-determining factor may represent a general principle to ensure stability in cellular differentiation. Finally, since Treg cells are widely pursued as an option for adoptive transfer-mediated cellular therapy, our finding offers a potentially novel therapeutic target to ensure long-term stability of in vitro-expanded patient-derived iTreg cells for clinical applications.

## Methods

**Mice.** Foxp1^{f/f} mice were generated in the Rudensky laboratory in University of Washington, Seattle. Foxp3^{IRES-YFP-Cre} mice[23], Foxp3^{eGFP-Cre-ERT2}R26Y mice[26], Foxp3^{GFPKO} mice[25], Foxp3^{GFP} mice[49], and Foxp3^{Thy1.1} mice[20] are described elsewhere. Vav-Bcl2^{tg} mice[50] was a gift from Dr. Jerry M. Adams at the Walter and Eliza Hall Institute of Medical Research, Australia. Smad2^{f/f}Smad3^{+/−}CD4-Cre mice[12] was a gift from Dr. Akihiko Yoshimura, Keio University School of Medicine, Japan. All mice except Smad2^{f/f}Smad3^{+/−}CD4-Cre mice were generated on or backcrossed to C57BL/6 background or C57BL/6 with CD45.1 allele wherever applicable. Smad2^{f/f}Smad3^{+/−}CD4-Cre mice are in C57BL/6 and 129β mixed background. All animals were maintained in specific pathogen-free barrier facilities and were used in accordance with protocols approved by the POSTECH Institutional Animal Care and Use Committee.

**FACS staining and antibodies.** For surface staining, fluorochrome-conjugated antibodies were purchased from eBioscience, Biolegend, Tonbo, and R&D Biosciences. Anti-CD4 (GK1.5), anti-CD8 (53−6.7), anti-CD25 (PC61.5), anti-CD24 (M1/69), anti-CD103 (2E7), anti-GITR (DTA-1), anti-ICOS (C398.4A), anti-CTLA4 (UC10-4B9), anti-Ki67 (16A8), anti-CD62L (MEL-14), anti-CD44 (IM7), anti-CD45.2 (104), anti-CD45.1 (A20), anti-Nrp1 (3DS304M), anti-TCRβ (H57-597), and anti-Thy1.1 (OX-7) were used at 1:400 dilution. Anti-Nrp1 (3DS304M) was used at 1:25 dilution.. For intracellular staining, surface-stained cells were fixed and permeabilized with a Foxp3 Staining Kit (eBioscience) according to the manufacturer's instruction and were stained with the following antibodies at 1:200 dilution: anti-Foxp3 (FJK-16s), anti-Helios (22F6), anti-IFN-γ (XMG1.2), anti-IL-17A (eBio17B7), anti-IL-4 (11B11), and anti-IL-13 (eBio13A) wherever applicable. For intracellular staining of Foxp1, cells were washed, fixed, and permeabilized as described and then incubated with anti-Foxp1 (Cell Signaling Technologies (CST) #2000S) antibody at 1:250 dilution. Subsequently, they were washed and further stained with anti-Rabbit IgG (CST-Poly4064) at 1:500 dilution followed by intracellular staining of Foxp3 and other surface markers. To reduce background, Fc Blocker was used before each staining step. For in vitro cultures, cells were stained first with Ghost Violet 510 Viability Dye prior to surface or intracellular staining at 1:1000 dilutions. PE Annexin V Apoptosis Detection Kit (BD Pharmingen™) was used for viability assay of cells cultured for in vitro iTreg induction. Stained cells were analyzed in LSR Fortessa SORP flow cytometer with the DIVA software (BD Biosciences) and were analyzed by FlowJo (Treestar). Gating strategies for FACS sorting and analyses are described in Supplementary Fig. 12 and 13, respectively.

**Histology.** Mouse tissues were fixed in 10% neutral-buffered formalin and were further processed by routine staining with hematoxylin and eosin. Blinded assessment of at least three sections of each tissue per mouse was performed by expert pathologist. Histological assessment of colon injury was performed at ×100 as previously described[51]. Histology score was generated on a scale from 0 to 4 as follows: 0, normal tissue; 1, inflammation with scattered cellular infiltration and no signs of epithelial degeneration; 2, moderate inflammation with multiple foci and/or mild epithelial ulcerations; 3, severe inflammation with marked wall thickening and/or ulcerations in >25% of the tissue; and 4, inflammation with inflammatory cell infiltration and/or >75% of the tissue section affected. An average of at least three fields of view per colon was evaluated blindly for each mouse.

**In vitro iTreg differentiation.** CD4⁺ cells were prepared from Foxp1^{+/+}Foxp3^{IRES-YFP-Cre} or Foxp1^{f/f}Foxp3^{IRES-YFP-Cre} by negative selection of total lymphocytes from spleen and peripheral lymph nodes using MACS LD columns (Miltenyi Biotec). The enriched cells were sorted using MOFLO ASTRIOS Flow Cytometer System (Beckman Coulter) to isolate naive CD4⁺CD62L^{hi} (YFP⁻ for Foxp3^{IRES-YFP-Cre} reporter mice) T cells from the total CD4⁺ cells. The purity of sorted T cell populations routinely exceeded 99%. These cells were activated with plate-bound anti-CD3(1 μg/ml) and anti-CD28 (1 μg/ml) in the presence of 2 ng/ml premium grade human TGF-β (Miltenyi Biotec) and 50 IU/ml of recombinant human IL-2 (Miltenyi Biotec) and cultured in complete RPMI1640. For prolonged cultures till 6–8 days, cells were split daily, starting from 2 days after TGF-β treatment in complete RPMI1640 supplemented with 50 IU/ml IL-2.

**Adoptive transfer and in vivo Treg cell maintenance assay.** Sorted CD45.2⁺CD4⁺Foxp3⁻CD62L^{hi} Tnv cells from WT and KO mice were mixed with sorted CD45.1⁺ Treg cells (1:3) and adoptively transferred intravenously (i.v.) to RAG1⁻/⁻ host mice. After 2 months of transfer, mice were analyzed to determine Foxp3⁺ cells within CD45.2⁺ population. For in vivo iTreg maintenance assay, 2 × 10⁵ FACS-sorted in vitro-generated iTreg cells from day 3 culture were co-injected with 6 × 10⁵ CD4+ T cells enriched from CD45.1⁺Foxp3^{GFPKO} mice into RAG1⁻/⁻ recipient mice. For an analogous experiment with thymic Treg cells, CD4SPFoxp3⁺ or CD4⁺Foxp3⁺CD24^{hi} or CD24^{lo} Treg cells were sorted from thymus of WT or KO mice wherever applicable. Thymic Treg cells were mixed with CD4⁺ T cells enriched from CD45.1⁺Foxp3^{GFPKO} cells at 1:3 ratio and i.v. injected to RAG1⁻/⁻ host mice.

**In vivo Treg fate mapping experiments.** CD45.2⁺CD4⁺CD25⁻GFP⁻CD62L^{hi} Tnv cells (with >99.7% purity) from Foxp1^{f/f}Foxp3^{GFP-Cre-ERT2}R26Y or Foxp1^{+/+}Foxp3^{GFP-Cre-ERT2}R26Y mice were transferred together with sorted CD45.1⁺Foxp3⁺ Treg cells in RAG1⁻/⁻ hosts who received a single oral gavage of tamoxifen (Sigma) in soybean oil (8 mg/20 g of body weight) 1, 2, or 5 weeks after cell transfer. Four weeks after treatment, recipient mice were sacrificed and YFP⁺ cells were analyzed. For flow cytometric analysis, CD4⁺ cells were enriched from spleen and peripheral lymph nodes and subjected to a staining protocol adapted from ref. [27]. Briefly, cells were pre-fixed with 0.5% paraformaldehyde for 10 min in order to retain the tamoxifen-induced YFP signal, before they were subjected to intracellular staining for Foxp3.

**Retroviral transduction.** Foxp1A cDNA was subcloned in MSCV-IRES-EGFP (MigR1-GFP) vector and transiently transfected into retroviral packaging cell line Platinum-E, using Fugene HD Transfection Reagent (Promega). For transduction, CD4⁺CD62L^{hi}Thy1.1⁻ Tnv cells from Foxp3^{Thy1.1} mice were sorted and activated

with plate-bound anti-CD3 and anti-CD28 overnight, followed by spin-infection with viral supernatant that was supplemented with HEPES and polybrene.

**Chromatin immunoprecipitation.** ChIP for Foxp1 was performed using a protocol developed by Upstate Biotechnology (17-295). Briefly, chromatin was prepared from $5 \times 10^6$ cells either sorted from mice or in vitro differentiated at the indicated time points and subjected to immunoprecipitation overnight at 4 °C with rabbit anti-Foxp1 (CST, D35D10 XP, #4402) or normal rabbit IgG (CST, #2729 S) from Cell Signaling Technology. Crosslinks were reversed and enrichment of the indicated DNA sequences were determined by quantitative real-time PCR with a SYBR Green (USB Veriquest™ Fast SYBR qPCR master mix with ROX) followed by quantification with the ViiA™7 Software v1.2.2. Within the regions of interest, Foxp1 binding is represented as the percentage of enrichment over input, and calculation was conducted using the change-in-threshold (2^{-ΔΔCT}) method[52]. To determine the status of histone modifications, ChIP analysis was performed as described elsewhere[13]. Antibodies specific for different histone modifications were from Abcam ((anti-H3K9me3(ab8898), anti-H3K4me3(ab8580), and Thermo Fisher Scientific (anti-H3K9/14/Ac (49-1010)). The sequences for primer pairs used for ChIP–quantitative PCR analyses are mentioned in supplementary table 1.

**Gene expression analysis.** Total RNA was isolated from sorted in vitro-induced iTreg cells populations using TRIZOL reagent (Ambion) and quantified in Nanodrop2000c (Thermo Fischer), and cDNA was reverse transcribed using Quantitect reverse transcription kit (QIAGEN) following the manufacturer's instruction. Quantitative real-time PCR was done with Fast SYBR Green (USB Veriquest™ Fast SYBR qPCR master mix with ROX) with the set of primers mentioned in supplementary table 1. A ViiA7 Real-Time PCR system (Life Technologies) was used for all reactions and detection. Quantification of relative mRNA expression was normalized to the expression of the control gene encoding *Hprt* and relative expression was calculated by change-in-threshold method.

**Adoptive-transfer-induced colitis.** Splenic CD4+Foxp3−CD45RBhi cells were sorted from *Foxp3*GFPCD45.1+ mice and mixed with total wild-type nTreg cells or Treg cells sorted directly from the thymus of *Foxp1*+/+*Foxp3*IRES-YFP-Cre or *Foxp1*f/f*Foxp3*IRES-YFP-Cre mice. For iTreg co-transferred groups, Treg cells were first induced in vitro from *Foxp1*+/+*Foxp3*IRES-YFP-Cre- or *Foxp1*f/f*Foxp3*IRES-YFP-Cre-derived naive T cells, and YFP+ iTreg cells were sorted after 3 days of induction. Colitis was induced in RAG1-deficient hosts by i.v. injection of $7.5 \times 10^5$ CD45RBhi splenocytes along with $2.5 \times 10^5$ indicated Treg cells. Mice weight was recorded blindly once weekly for 8 weeks. Colon were isolated from hosts and weighed at the end of the experiment.

**DSS-induced colitis.** Chronic colitis was induced by the provision of 2% (wt/vol) DSS (molecular mass, 36–50 kilodaltons; MP Biomedicals) in drinking water ad libitum. Two cycles of DSS and 2 cycles of normal drinking water (7 days/cycle) was provided alternatively. Mice weight was recorded blindly from day 1 of DSS through day 25 of treatment. At the end of the experiment, mice were sacrificed by $CO_2$ asphyxiation and colon were removed and lymphocytes were isolated for FACS analyses. Horizontal tissue sections from colon were also fixed for further histopathological analyses.

**Bisulfite sequencing.** Total genomic DNA from in vitro-induced Treg cells at day 7 in culture was isolated using the Accuprep Genomic DNA Extraction Kit (Bioneer). Bisulfite conversion of DNA was performed with the MethylEasy Xceed Kit (Human Genetic Signature). PCR amplicon from loci of interest was generated from this treated DNA using primers mentioned in supplementary table 1. After electrophoresis to confirm the size, these PCR products were then purified from agarose gel using HiGene Gel and PCR purification system (Biofact). Library preparation followed by high-throughput sequencing (150 bp, paired-end) of the DNA was performed by Theragen Etex (Republic of Korea). High-throughput sequencing (150 bp, paired-end) of the prepared DNA libraries was performed using Illumina HiSeq4000. The sequenced reads were trimmed by quality (Phred quality score cutoff 20) using Trim Galore (version 0.4.2, https://www.bioinformatics.babraham.ac.uk/projects/trim_galore/) with Cutadapt (version1.12) (ref. http://journal.embnet.org/index.php/embnetjournal/article/view/200/479). The trimmed reads were mapped to amplicon sequences using Bismark (version 0.17.0)[53]. The ratio of methylated and unmethylated CpG at each CpG site was calculated from the Bismark analysis.

**In vitro suppression assay.** Sorted Nrp1+YFP+ and Nrp1−YFP+ cells from *Foxp1*+/+*Foxp3*IRES-YFP-Cre or *Foxp1*f/f*Foxp3*IRES-YFP-Cre mice were incubated with responder cells (CD4+Foxp3− CD62Lhi) that were pre-pulsed with CTV for 10 min at 37 °C. Cells were washed in phosphate-buffered saline twice and immediately used. T-cell-depleted splenocytes ($1 \times 10^5$ cells) were mixed with CTV-pulsed responder cells ($5 \times 10^4$) and the indicated amounts of Treg cells of each group along with 0.3 μg/ml of anti-CD3 and plated in round-bottom 96-well plate. Cells were cultured for 4 days and their proliferation was analyzed by flow cytometry for determining the dilution and mean fluorescent intensity of CTV intensity.

**Statistics.** Statistical analysis was performed using Graphpad Prism 6.0 (Graph Pad software, La Jolla, CA, USA) operating unpaired two-tailed Student's *t* test or two-way analysis of variance with Bonferroni post-test where applicable. Statistically significant differences in mean values were considered at $P < 0.05$.

## Data availability

All data that support the findings of this study are available from the corresponding author upon request. The bisulfite sequencing data were deposited in the Gene Expression Omnibus (NCBI) data repository under accession number GEO: GSE118595.

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

## Acknowledgements

We thank Dr. Li-fan Lu, UCSD for suggestions and critical reading of the manuscript. We also thank Haejin Jung for technical assistance for cell sorting. This work was supported by Project IBS-R005 from the Institute for Basic Science, Korean Ministry of Science, and ICT.

## Author contributions

S.G. designed and performed the experiments, analyzed data, and wrote the manuscript; S.R.-C. performed histopathology analyses; K.K. performed computational analysis of bisulfite sequencing; S.-H.I. provided intellectual suggestions during the course of the study and edited the manuscript; D.R. directed the project, was involved in designing and analyses of experiments, and wrote the manuscript.

## Additional information

**Competing interests:** The authors declare no competing interests.

