## [Peer Review file · Nature Communications]

Reviewers' comments:

Reviewer #1 (Immune regulation, Treg)(Remarks to the Author):

Ghosh et al. The transcription factor, FoxP1 preserves integrity of an active FoxP3 locus in extrathymic Tregs.

This manuscript shows that Tregs with a FoxP3-inducible deletion of FoxP1 lose FoxP3 expression over time. The loss of FoxP3 expression occurs in vitro in TGFb-induced Tregs as well as Tregs that have been transferred in vivo. Only inducible Tregs are affected, as thymic Tregs with FoxP1 deletion maintain their expression of FoxP3. Importantly, FoxP1 binds to the promoter and CNS2 regions of the FoxP3 locus and its binding is dependent on TGFb signaling through active SMAd2/3. Although FoxP1 does not appear to impact FoxP3 locus DNA methylation, it strongly affects H3-K4me3 and H3-K9/14Ac marks around the transcriptional start site of the FoxP3 locus. In mice that lack expression of FoxP1 in FoxP3-expressing cells, the Nrp1+Helios+ thymic Treg population is increased in frequency and the Nrp1-Helios- iTregs are decreased in frequency. In addition, IL-17-expressing CD4+ T cells are modestly increased in aged mice – a change that correlates with increased intestinal pathology.

The data convincingly demonstrate that, particularly for TGFb-induced iTregs, FoxP1 is important for acutely maintaining FoxP3 expression. Moreover, this effect seems to occur through FoxP1-mediated transcriptional changes at the FoxP3 locus. However, the in vivo impact of these changes is less clear.

1. In figure 6, most of the data report frequencies of various Treg populations, but fail to report the actual numbers. The numbers/organ are essential for understanding what is happening in vivo. Numbers are also important for understanding the KO-Tg cells that express Bcl2.
2. Despite the loss of FoxP1, many Nrp1-Helios- iTregs remain. Do these cells have reduced FoxP3 expression (mRNA or protein)? What about the Nrp1+Helios+ tTregs?
3. Is the suppressive function of either population affected by the loss of FoxP1?
4. What is the fate of T cells that have a FoxP3-specific deletion of FoxP1? Do they become Th17 cells? This seems like a critical point as the authors generated in Fig 2 FoxP3-expressing cells that are permanently marked with YFP, many of which later lose FoxP3 expression. What are they?
5. For the remainder of the FoxP1-deleted cells that retain expression of FoxP3, are they still suppressive?

Reviewer #2 (Treg, epigenetic/DNA methylation)(Remarks to the Author):

Ghosh et al examined the role of Foxp1 expression on the stability of Foxp3 expression in peripherally induced regulatory T cells (iTreg). They found that Foxp1 overexpression stabilizes Foxp3 in iTreg, while its deletion destabilizes Foxp3 in vitro and in vivo in iTreg, but not in thymus-derived Treg (tTreg). Foxp1-deficient iTreg seem to cause increased intestinal inflammation (but only in aged mice) and aggravated colitis in the adoptive transfer model.

Overall, the laborious experiments are well planned and performed with high technical expertise. The main caveat of the manuscript is the lack of a convincing phenotype in mice knocked-out for Foxp1 in iTreg. Therefore, doubts remain on the physiological relevance of Foxp1 expression in iTreg.

Minor comments:

Text: It is a long stretch with several supplemental Figures shown before coming to the results in Fig. 1. Throughout the manuscript the reader continuously needs to switch back and forth between figures and supplemental Material. This could be improved.

Figure. 2: Very low frequencies of iTreg re-isolated from LN and spleen are shown. As the authors point out that iTreg generation occurs primarily in the GI tract it would be nice to also show data from large and/or small intestine, if available.

Fig. 5: Foxp1 binds to the Foxp3 promotor and CNS2, but there seems no influence of CpG methylation on Foxp1 binding, but lower permissive trimethylation modifications of H3-K4me3 and H3-K9/14Ac at d7 in culture were observed. Overall, the link between Foxp1 binding and the epigenetic changes remains unclear.

Fig. 6: 1st sentence of legend 6a is wrong, please correct.

Suppl. Fig. 9a: the increase in cellularity suggested in text line 335 is not really convincing in this graph, similarly, the influence of the Foxp1 ko on cytokine secretion and T cell activation in tissues seems not impressive. Same applies to Fig. 6f and suppl. Fig. 10.

Fig. 6g: please provide summarized data and histological scoring.

Suppl. Fig. 11: Considering the instability of in vitro induced iTreg the frequency of Foxp3 in the inoculum could be described. Overall, this experiment is not fully convincing in the clarification of the role of Foxp1 for iTreg-mediated suppression of colitis. As in vitro induced wt iTreg protected from colitis it would have been nice to use the tamoxifen system for the timed ko of Foxp1 in this setting. In the legend of suppl. Fig. 11 it is stated that 13 in vitro induced iTreg cell products were transferred into RAG ko recipients. Please show all summarized data in the Figure.

A point-by-point response to reviewers' comments

In the beginning we want to thank the reviewers for their time to carefully scrutinize our findings and provide us the opportunity to submit a revised version of our manuscript NCOMMS-18-10393 entitled "The transcription factor Foxp1 preserves integrity of an active *Foxp3* locus in extrathymic Treg cells". We have taken into account all the reviewers' concerns and performed further experiments to address those. Following reviewers' suggestions, we have also clarified and discussed some of the data presented in the manuscript, and we have modified the text where necessary. Please, find below a detailed point-by point reply to the reviewers.

Reviewer 1

General comment:

Ghosh et al. The transcription factor, FoxP1 preserves integrity of an active FoxP3 locus in extrathymic Tregs.

This manuscript shows that Tregs with a FoxP3-inducible deletion of FoxP1 lose FoxP3 expression over time. The loss of FoxP3 expression occurs in vitro in TGFb-induced Tregs as well as Tregs that have been transferred in vivo. Only inducible Tregs are affected, as thymic Tregs with FoxP1 deletion maintain their expression of FoxP3. Importantly, FoxP1 binds to the promoter and CNS2 regions of the FoxP3 locus and its binding is dependent on TGFb signaling through active SMAd2/3. Although FoxP1 does not appear to impact FoxP3 locus DNA methylation, it strongly affects H3-K4me3 and H3-K9/14Ac marks around the transcriptional start site of the FoxP3 locus. In mice that lack expression of FoxP1 in FoxP3-expressing cells, the Nrp1+Helios+ thymic Treg population is increased in frequency and the Nrp1-Helios-iTregs are decreased in frequency. In addition, IL-17-expressing CD4+ T cells are modestly increased in aged mice – a change that correlates with increased intestinal pathology.

The data convincingly demonstrate that, particularly for TGFb-induced iTregs, FoxP1 is important for acutely maintaining FoxP3 expression. Moreover, this effect seems to occur

through FoxP1-mediated transcriptional changes at the FoxP3 locus. However, the in vivo impact of these changes is less clear.

Response to general comment:

We appreciate the reviewer's comment regarding the *in vivo* impact of Treg specific deletion of Foxp1. To address this issue, we have employed an experimental model of colitis and observed that the *Foxp1^{fl/fl}Foxp3^{JRES-YFP-Cre}* (KO) mice are considerably more susceptible to disease severity by Dextran Sodium Sulfate (DSS) induced colitis compared to their WT littermates. In accordance to the moderate age related inflammation at the mucosal site of GI tract, these mice even at a young age (6 to 8 weeks) responded to DSS induced colitis with increased body weight loss subsequent to having higher percentage of IL17⁺ and IFN γ ⁺IL17⁺ effector T cell populations and extensive cellular infiltration, erosion and edema at the mucosal surface of colon. We have included this data in the new Fig. 7d-f. The text describing the experiment is highlighted and corresponds to lines 424-432 of the revised manuscript. Also we have moved Fig. 6e-g from our initial submission to Fig. 7a-c, to separate the overall characterization of lymphoid organs (revised Fig. 6a-d) and the intestinal phenotype of these mice. We believe this would make the flow of the manuscript more comprehensible for the readers.

Specific comment 1:

In figure 6, most of the data report frequencies of various Treg populations, but fail to report the actual numbers. The numbers/organ are essential for understanding what is happening in vivo. Numbers are also important for understanding the KO-Tg cells that express Bcl2.

Response to specific comment 1:

We thank the reviewer for this insightful suggestion. Accordingly we have calculated the actual cell numbers with the corresponding percentage of various Treg populations both in *Foxp3^{JRES-YFP-Cre}* and *Foxp3^{JRES-YFP-Cre} Bcl2^{Tg}* mice. It is evident from the data that while unlike percentage, the total cell numbers of the populations of Treg cells expressing these activation and proliferation markers is only marginally affected in the lymphoid organs of KO mice, the Nrp1⁻ Helios⁻ Treg population is the primary population that is most significantly decreased in terms of

both frequency and absolute numbers. The data is incorporated in the new version of Fig. 6a-c and Supplementary Fig. 9c, d. Corresponding texts are highlighted in lines 326, 331-332, 337-338.

Specific comment 2:

Despite the loss of FoxP1, many Nrp1-Helios- iTregs remain. Do these cells have reduced FoxP3 expression (mRNA or protein)? What about the Nrp1+Helios+ iTregs?

We thank the reviewer for suggesting this experiment, since the results revealed interesting aspects about the residual iTreg population in the *Foxp1^{fl/fl} Foxp3^{JRES-YFP^{cre}}* (KO) mice that we didn't investigate before. Staining with both Helios and Nrp1 in order to sort out the Helios⁻Nrp1⁺ Treg cells involves fixation and permeabilization and would preclude us from using live cells for suppression assay (response to specific comment 3, below). Therefore for this experiment we used only Nrp1^{lo} population as the marker of iTreg cells, which were almost equally affected in these mice (Supplementary Fig. 7b), and are also routinely referred to as peripherally induced Treg cells^{1,2,3}. Interestingly, by real time PCR as well as intracellular staining, we found that while Foxp1 was almost completely deleted in the Nrp1^{hi} tTreg population, approximately 8-10% of the residual Nrp1^{lo} Treg cells derived from the KO mice still retained Foxp1. Furthermore, post sort staining and real time PCR showed no difference in Foxp3 expression between WT and KO Nrp1^{lo} cells. This observation suggested to us that these cells represent a small population of recently generated iTreg cells in KO mice that are still in the process of completely deleting Foxp1, and extended the possibility that giving them more time would lead to complete deletion of Foxp1 and eventual instability in Foxp3. Indeed when cultured *in vitro* along with allelically marked Tnv cells for 3 days, these cells displayed complete loss of Foxp1 and more importantly demonstrated instability in Foxp3 expression. This experiment is described in the new Supplementary Fig. 8a-g, and highlighted text, lines 342-361.

Specific comment 3:

Is the suppressive function of either population affected by the loss of FoxP1?

Response to specific comment 3:

Following reviewer's suggestion in connection to the previous experiment we also performed *in vitro* suppression assay with the sorted Nrp1^{hi} and Nrp1^{lo} Treg cells from both WT and KO mice. As shown in the new Supplementary Fig. 8h, while the Nrp1^{hi} populations were comparable, the Nrp1^{lo} Treg population from the KO mice displayed marginally reduced suppressive activity (highlighted text in lines 362-364).

Specific comment 4:

What is the fate of T cells that have a FoxP3-specific deletion of FoxP1? Do they become Th17 cells? This seems like a critical point as the authors generated in Fig 2 FoxP3-expressing cells that are permanently marked with YFP, many of which later lose FoxP3 expression. What are they?

Response to specific comment 4:

This is indeed a very appropriate question and we completely agree with the reviewer that it is essential to know the phenotype of the Foxp1 deleted iTreg cells that lost Foxp3 expression. In the experimental setting described in Fig. 2, the number of YFP marked cells generated after Tamoxifen treatment is very less. Furthermore a fraction of them lose Foxp3 upon deletion of Foxp1. This very limited cell number restricts meaningful analyses of cytokine production from these cells after stimulation.

To circumvent this problem we employed a modified version of the experimental scheme where we pulsed naïve T cells sorted from *Foxp1^{fl/fl} Foxp3^{eGFP-Cre-ERT2} R26Y* or control *Foxp1^{+/+} Foxp3^{eGFP-Cre-ERT2} R26Y* mice with TGFβ for 24 hours before transferring to RAG1 deficient hosts. The recipients received Tamoxifen after 2 days of cell transfer and were analyzed 7 days thereafter. As expected, this experimental set up enabled convenient analysis as a reasonably larger YFP⁺ population was generated. In the absence of Foxp1, the YFP⁺Foxp3⁻ population was found to produce significantly higher amount of IFNγ and showed a higher trend of IL17 production, while other cytokines analyzed, IL2 or IL4, remained unchanged. We have included this data in the new Fig. 2e-f. The text describing the experiment is highlighted and corresponds to lines 202-218 of the revised manuscript.

Specific comment 5:

For the remainder of the FoxP1-deleted cells that retain expression of FoxP3, are they still suppressive?

Response to specific comment 5:

While we agree with the reviewer that it would be interesting to know the suppressive capacity of the YFP⁺Foxp3⁺ cells in the fate mapping experiment, we would like to humbly point out the following issues that made it technically very challenging to perform this experiment. For analyzing Foxp3⁺ and Foxp3⁻ cells among YFP⁺ population, the cells were briefly fixed with formaldehyde followed by Foxp3 staining. Since the cells are dead after fixation, this precludes us from using this method of distinguishing YFP⁺Foxp3⁺ and YFP⁺Foxp3⁻ populations for suppression assay. Furthermore, since Foxp3 allele is equipped with eGFP-Cre-ERT2 in these fate-mapping mice, GFP and YFP could theoretically be used to distinguish between these two populations by using specific lasers. However our attempt to do so yield literally few hundred cells with insufficient purity, most likely due to the extremely low frequency of the YFP⁺ population that still retained GFP.

Please note, in Fig. 2c, where cells were sorted back from RAG1^{-/-} recipient mice to confirm efficient deletion of Foxp1, we had similar technical problem and found it difficult to get enough cells even for staining. Therefore total cells in the CD4⁺FITC⁺ gate were sorted out that contained both CD45.2⁺YFP⁺ as well as CD45.1⁺Foxp3^{GFP+} cells. CD45.1⁺ cells were gated out in order to assess Foxp1 deletion.

Reviewer 2

General comment:

Ghosh et al examined the role of Foxp1 expression on the stability of Foxp3 expression in peripherally induced regulatory T cells (iTreg). They found that Foxp1 overexpression stabilizes Foxp3 in iTreg, while its deletion destabilizes Foxp3 in vitro and in vivo in iTreg, but not in thymus-derived Treg (tTreg). Foxp1-deficient iTreg seem to cause increased intestinal

inflammation (but only in aged mice) and aggravated colitis in the adoptive transfer model. Overall, the laborious experiments are well planned and performed with high technical expertise. The main caveat of the manuscript is the lack of a convincing phenotype in mice knocked-out for Foxp1 in iTreg. Therefore, doubts remain on the physiological relevance of Foxp1 expression in iTreg.

Response to general comment:

We really appreciate the reviewer's comment. In our initial study we established that the absence of Foxp1 selectively compromise iTreg homeostasis in *Foxp1^{fl/fl} Foxp3^{RES-YFP-cre}* mice at steady state condition, leading to unprovoked colonic phenotype at an older age. In light of these findings, in order to more directly investigate the physiological relevance of Foxp1 deletion in iTreg cells, we investigated whether this steady state immunological imbalance might trigger these mice to respond to Dextran Sodium Sulfate (DSS) induced colitis with exacerbated disease severity. As shown in newly introduced Fig. 7d-f the KO mice exhibited more severe disease progression as manifested by significantly enhanced body weight loss, higher percentage of IL17⁺ and IFN γ ⁺IL17⁺ effector T cell populations in the large intestine and higher histopathological scores as obtained from colonic tissues compared to WT littermate controls. Thus, Treg specific ablation of Foxp1 resulted in severely reduced iTreg compartment in lymphoid organs and gut associated mucosal sites, which contributed to unprovoked age related mucosa associated inflammation as well as enhanced susceptibility to a physiologically relevant experimental model of colitis even at a young age.

The experiment is described in highlighted lines 424-432 of the revised manuscript. Please note we have also moved Fig. 6e-g from our initial submission to Fig. 7a-c since we believed it would be more appropriate to separate characterization of lymphoid organs (revised Fig. 6a-d) and the intestinal phenotype of these mice, in order to blend in with the flow of the manuscript.

Minor comment 1:

Text: It is a long stretch with several supplemental Figures shown before coming to the results in Fig. 1. Throughout the manuscript the reader continuously needs to switch back and forth between figures and supplemental Material. This could be improved.

Response to minor comment 1:

We are extremely sorry for making the reviewer confused. In the revised manuscript we have merged Supplementary Fig. 2 and Fig. 1 from our initial submission to make the new Fig. 1. We hope that this new arrangement will substantially reduce the readers' effort from moving back and forth between main Fig. 1 and Supplementary Fig. 2.

Minor comment 2:

Figure 2: Very low frequencies of iTreg re-isolated from LN and spleen are shown. As the authors point out that iTreg generation occurs primarily in the GI tract it would be nice to also show data from large and/or small intestine, if available.

Response to minor comment 2:

We understand the reviewer's concern about the issue of cell number. Since in this experiment the YFP⁺ population generated in the lymphoid organs upon Tamoxifen treatment was substantially low, it was technically very challenging to probe the cells from single cell suspension of individual lymphoid organs. We therefore, pooled peripheral lymph node, mesenteric lymph node and spleen and enriched CD4⁺ cells by MACS magnetic enrichment protocol from each host separately. Therefore for the complete analyses we performed CD4 enrichment on total 26 individual samples (13 each group), ~ 8-10 samples per experimental replicates. Unfortunately these analyses used to be such rigorous and intensive that we could not undertake T-cell isolation from GI tract lamina propria at the same time from the same host. Hence, we deeply apologize for not having the GI tract data available for these experiments.

Minor comment 3:

Fig. 5: Foxp1 binds to the Foxp3 promotor and CNS2, but there seems no influence of CpG methylation on Foxp1 binding, but lower permissive trimethylation modifications of H3-K4me3

and H3-K9/14Ac at d7 in culture were observed. Overall, the link between Foxp1 binding and the epigenetic changes remains unclear.

Response to minor comment 2:

We agree with the reviewer that the connection between Foxp1 mediated chromatin modification and its relationship with CpG methylation is not entirely clear. However, given that our understanding about the sequence of epigenetic events contributing to gene expression are still evolving, to what extent specific chromatin modifications are related to DNA hypo or hypermethylation, and whether it differs for different genetic loci, is not yet completely well characterized. To our understanding, under certain circumstances, hypomethylation of DNA and chromatin remodeling acts at different layers of genome modification for permitting gene expression in the vicinity. For example, in agreement to our finding, it is reported using mouse embryonic stem cells (mESCs), that the active histone modification mark H3K4Me3 is indeed deposited independent of DNA methylation⁴. Furthermore, in the context of iTreg cells, TGF β signaling has been shown to upregulate H3K4 trimethylation at the *Foxp3* locus, keeping the DNA methylation status unaltered^{5,6}. Keeping these points in consideration, we have modified the discussion section, and added the above references (highlighted text, lines 473-481, 938-940 and 979-983). We believe that TGF β signaling in the context of epigenetic modifications in the *Foxp3* locus, and mechanistic contribution of Foxp1 in the whole process is worth pursuing as a model system for in depth understanding of basic aspects of epigenetics and its connection to gene expression. We will be very interested to pursue that in the future.

Minor comment 4:

Fig.6: 1st sentence of legend 6a is wrong, please correct.

Response to minor comment 4:

We realize that we have not indicated the specific lymphoid organs and the specific gate used for the analysis. We have updated the sentence by providing this information. (Highlighted texts, line 633-635).

Minor comment 5:

Suppl. Fig. 9a: the increase in cellularity suggested in text line 335 is not really convincing in this graph, similarly, the influence of the Foxp1 ko on cytokine secretion and T cell activation in tissues seems not impressive. Same applies to Fig. 6f and suppl. Fig. 10.

Response to minor comment 5:

We are extremely sorry for not being clear in the text. Since the increase in cellularity was significantly higher only in the lymph nodes, we stated “Subsequent analyses of the KO mice displayed increased cellularity compared to WT, specifically within lymph nodes...” (Lines 334-335 in the original submission). We have modified this sentence by saying “Subsequent analyses of the KO mice displayed moderate increase in cellularity specifically within lymph nodes compared to WT counterparts...” (Highlighted text, lines 378-379 in the current version).

In case of cytokine secretion within GALT, we observed specific increase in IL17 producing T cells only in large intestine of aged mice. We therefore presented the data for the cytokine producing cells, CD4⁺IFN γ ⁺, CD8⁺IFN γ ⁺ and CD4⁺IL4⁺ that were not affected, side by side for young and old mice in Supplementary Fig. 10. Furthermore in the main figure (previous Fig. 6f) we presented representative histogram and statistics only for IL17, which was significantly increased specifically within the large intestine of aged mice. In order to make our statement more clear, we modified our previous sentence “While GALT tissue derived T-cells displayed no detectable differences in terms of pro-inflammatory cytokine expression at young age, there was a clear trend of increased frequencies of IL17 producing T-cells most prominently in the LI-LP of aged mice (Fig. 6f and Supplementary Fig.10)” (lines 345-348 in the original submission). The new sentences read “The GALT tissue derived T-cells displayed no detectable differences in terms of pro-inflammatory cytokine expression at young age (Fig. 7b, top left and middle panel, Supplementary Fig. 10 left panel). However, there were significantly increased frequencies of IL17 producing T-cells specifically within the LI-LP of aged mice (Fig. 7b, top right and lower panel, Supplementary Fig. 10 right panel)”. (Highlighted text, lines 392-397 in the current version).

Minor comment 6:

Fig. 6g: please provide summarized data and histological scoring.

Response to minor comment 6:

We have provided the summarized data of histological scoring in the new Fig. 7c, which is the current position of the old Fig. 6g. The corresponding texts are highlighted in line 399 and method for histological score calculation is included in “Material and Methods” section (lines 703-711 and 1009-1010 of the current manuscript).

Minor comment 7:

Suppl. Fig. 11: Considering the instability of in vitro induced iTreg the frequency of Foxp3 in the inoculum could be described. (a)

Overall, this experiment is not fully convincing in the clarification of the role of Foxp1 for iTreg-mediated suppression of colitis. As in vitro induced wt iTreg protected from colitis it would have been nice to use the tamoxifen system for the timed ko of Foxp1 in this setting. (b)

In the legend of suppl. Fig. 11 it is stated that 13 in vitro induced iTreg cell products were transferred into RAG ko recipients. Please show all summarized data in the Figure. (c)

Response to minor comment 7a:

We agree with the reviewer that since *in vitro* generated iTreg cells derived from KO mice gradually lose Foxp3; it would be helpful to comment on Foxp3 expression level in the co-transferred iTreg cells in the experiment corresponding to Supplementary Fig. 11. Since for this experiment we sorted iTreg cells after three days induction, at this early point when Foxp1 is not completely deleted, the expression of Foxp3 is similar for both WT and KO derived iTreg cells. The expression of Foxp3 in both groups at this point is essentially the same as what is shown in previous Fig. 1e, left panel and new Fig. 1h, left panel.

In order to make it clear for the reader, we have modified the text describing this part. The previous sentence “Colitogenic CD4⁺Foxp3⁻CD45RB^{hi} T-cells were transferred in RAG1^{-/-} recipients either alone, along with total WT nTreg cells, with sorted tTreg cells from the thymus of young WT or KO mice, or with 3 days *in vitro* differentiated iTreg cells derived from WT or KO mice” now reads “Colitogenic CD4⁺Foxp3⁻CD45RB^{hi} T-cells were transferred in RAG1^{-/-} recipients either alone, along with total WT nTreg cells, with sorted tTreg cells from the thymus of young WT or KO mice, or with 3 days *in vitro* differentiated iTreg cells derived from WT or KO mice. Of note, as shown earlier (Fig. 1h left panel) at this time point, both WT and KO derived iTreg cells retain comparable levels of Foxp3.” (Highlighted text, lines 406-408).

Response to minor comment 7b:

We agree with the reviewer that using the fate mapping mice would have been a cleaner setting for this experiment. However we would like to humbly point out that *in vivo* administration of Tamoxifen results in the deletion of conditional floxed alleles only in a fraction of the target cells, as is evident by our preliminary experiment presented in Rebuttal Fig. 1a. A single dose of Tamoxifen treatment only triggers ~10% of the total Treg population of the *Foxp3*^{eGFP-Cre-ERT2}*R26Y* mice to delete the Floxed stop allele of *R26Y* locus and express YFP. It is well established that even the presence of a fraction of wild type cells among functionally compromised total Treg population can prevent autoimmunity in mice. Therefore, in our effort to address reviewer’s concern, we reasoned that multiple doses of Tamoxifen treatment is required after co-transfer of iTreg cells generated from fate mapping mice, to efficiently delete *Foxp1* in majority of *Foxp1*^{fl/fl}*Foxp3*^{eGFP-Cre-ERT2}*R26Y* derived iTreg cells. Following experiment was performed:

Tnv cells were isolated from *Foxp1*^{fl/fl}*Foxp3*^{eGFP-Cre-ERT2}*R26Y* or control *Foxp1*^{+/+}*Foxp3*^{eGFP-Cre-ERT2}*R26Y* mice, iTreg cells were induced for three days and co-transferred with allelically marked CD4⁺Foxp3⁻CD45RB^{hi} cells in RAG1^{-/-} recipients. Three doses of Tamoxifen were administered as indicated in Rebuttal Fig. 1b. Unfortunately, all the groups including the positive control (CD45RB^{hi} + total nTreg) started losing weight, and died in the second week (Rebuttal Fig. 1c), presumably due to oestrogen independent toxicity from multiple dose of Tamoxifen treatment⁷. Therefore this experimental setting could not be employed to perform this experiment.

As an alternative approach, we treated the cells with 4-OH Tamoxifen *in vitro* at the time of iTreg induction, which was washed away after 48 hours. 24 hours after this, YFP⁺ iTreg cells were sorted and co-transferred in RAG1^{-/-} recipient mice along with colitogenic CD45RB^{hi} cells, and body weight of mice was monitored every week. In this setting we indeed observed enhanced weight loss in the experimental group that received KO Cre-ERT2 derived YFP⁺ iTreg cells, compared to controls. This data is presented in the new Supplementary Fig. 11b and corresponding highlighted text, lines 413-416.

Taken together we propose that this result, along with the data presented in Supplementary Fig. 11a, as well as the new data showing enhanced susceptibility of the KO mice to a DSS induced model of colitis strongly establish the phenotypic relevance of Foxp1 in Treg cell mediated immune tolerance in the intestine.

Response to minor comment 7c:

We apologize for not being clear. Although throughout the course of this experiment we co-transferred induced WT and KO derived iTreg cells into 13 recipient RAG1^{-/-} mice per group, for the KO group some mice died on the last week, and therefore the colon weights could not be taken. We have clarified this by modifying the legend of Supplementary Fig. 11. The first sentence now reads “Percentage body weight over time and colon weight of representative RAG1^{-/-} mice viable at the end of the experiment, which received....”.

Rebuttal Fig. 1

(a) Representative FACS plot showing percentage of YFP⁺ population converted from total gated CD4⁺ cells after single dose of TMX gavage. (b) Experimental scheme. Double FACS sorted >99% pure Tnv cells from control *Foxp1^{+/+}Foxp3^{eGFP-Cre-ERT2R26Y}* or *Foxp1^{fl/fl}Foxp3^{eGFP-Cre-ERT2R26Y}* mice were activated *in vitro* for 3 days with 1ug/ml of plate bound anti-CD3 and CD28 in the presence of TGFβ and 50 IU/ml of IL2. At day 3 GFP⁺CD25⁺ cells were FACS purified and co-transferred with sorted CD45RB^{hi} colitogenic cells to RAG^{-/-} host mice. Along with these two experimental groups there were two control groups which received either only CD45RB^{hi} cells or mixed with total WT nTreg cells. (c) Percentage body weight loss of the above mentioned groups only for one week; after which following 3rd TMX treatment all the animals died. (n ≥ 6 for all groups).

References

1. Weiss, J.M. *et al.* Neuropilin 1 is expressed on thymus-derived natural regulatory T cells, but not mucosa-generated induced Foxp3(+) T reg cells. *The Journal of Experimental Medicine* **209**, 1723-1742 (2012).
2. Yadav, M. *et al.* Neuropilin-1 distinguishes natural and inducible regulatory T cells among regulatory T cell subsets *in vivo*. *Journal of Experimental Medicine*, jem. 20120822 (2012).

3. Kim, K.S. *et al.* Dietary antigens limit mucosal immunity by inducing regulatory T cells in the small intestine. *Science* **351**, 858-863 (2016).
4. King, A.D. *et al.* Reversible Regulation of Promoter and Enhancer Histone Landscape by DNA Methylation in Mouse Embryonic Stem Cells. *Cell Rep* **17**, 289-302 (2016).
5. Ohkura, N. *et al.* T cell receptor stimulation-induced epigenetic changes and Foxp3 expression are independent and complementary events required for Treg cell development. *Immunity* **37**, 785-799 (2012).
6. Ohkura, N., Kitagawa, Y. & Sakaguchi, S. Development and maintenance of regulatory T cells. *Immunity* **38**, 414-423 (2013).
7. Huh, W.J. *et al.* Tamoxifen induces rapid, reversible atrophy, and metaplasia in mouse stomach. *Gastroenterology* **142**, 21-24 e27 (2012).

REVIEWERS' COMMENTS:

Reviewer #1 (Remarks to the Author):

The authors have provided additional data that appropriately address my original comments.

Reviewer #2 (Remarks to the Author):

All relevant issues have been addressed.

Response to Reviewers' comments:

Reviewer #1 (Remarks to the Author):

The authors have provided additional data that appropriately address my original comments.

Authors' response:

We really appreciate the reviewer's effort to critically read our manuscript and suggesting constructive experiments to improve our paper.

Reviewer #2 (Remarks to the Author):

All relevant issues have been addressed.

Authors' response:

We sincerely thank the reviewer for carefully examining our manuscript and approving the revised version for publication.